# Do Not Waste Your Rollouts: Recycling Search Experience for Efficient Test-Time Scaling

## Abstract

Test-Time Scaling enhances the reasoning capabilities of Large Language Models by allocating additional inference compute to broaden the exploration of the solution space. However, existing search strategies typically treat rollouts as disposable samples, where valuable intermediate insights are effectively discarded after each trial. This systemic memorylessness leads to massive computational redundancy, as models repeatedly re-derive discovered conclusions and revisit known dead ends across extensive attempts. To bridge this gap, we propose **Recycling Search Experience (RSE)**, a self-guided, training-free strategy that turns test-time search from a series of isolated trials into a cumulative process. By actively distilling raw trajectories into a shared experience bank, RSE enables positive recycling of intermediate conclusions to shortcut redundant derivations and negative recycling of failure patterns to prune encountered dead ends. Theoretically, we provide an analysis that formalizes the efficiency gains of RSE, validating its advantage over independent sampling in solving complex reasoning tasks. Empirically, extensive experiments on HMMT24, HMMT25, IMO-Bench, and HLE show that RSE consistently outperforms strong baselines with comparable computational cost, achieving state-of-the-art scaling efficiency.

## 1. Introduction

> *"Wisdom comes from experience. Experience is often a result of lack of wisdom."*
> — Terry Pratchett

Scaling parameters and data during pre-training has long been the primary driver of LLM performance, yet recent breakthroughs highlight a paradigm shift towards Test-Time Scaling (TTS) (Brown et al., 2024). By leveraging extended computation at test time, models can effectively tackle complex reasoning tasks that were previously out of reach (OpenAI, 2024; DeepSeek-AI et al., 2025). TTS approaches can be broadly divided into two categories: *Internal* TTS, which trains the LLMs to "think" slowly with long Chain-of-Thought (CoT) (Qwen Team, 2024; Kimi Team et al., 2025), and *External* TTS, which improves performance by allocating additional inference-time compute with a fixed LLM (Snell et al., 2024; Wu et al., 2025b). Focusing on the latter, test-time search has emerged as a prominent paradigm, where the model samples extensive rollouts to broaden the exploration of the solution space (Zhang et al., 2025; Liu et al., 2025).

While yielding promising performance gains, existing test-time search strategies share a common bottleneck in information utilization: rollouts are often treated as *disposable attempts* rather than *experience to be distilled and reused* (See Figure 1). Specifically, *parallel scaling* expands breadth via extensive independent rollouts, but typically offers limited cross-branch sharing of intermediate insights (Wang et al., 2023b; Lightman et al., 2024); *sequential scaling* iteratively improves a single draft, yet the information it accumulates is confined to unstructured in-context history: finite context windows force truncation or compression of earlier revisions, and the search remains a path-dependent local refinement around the current draft rather than a reusable pool of experience (Madaan et al., 2023; Shinn et al., 2023; Chen et al., 2024); *current hybrid scaling* uses external process reward models (Snell et al., 2024; Beeching et al., 2024; Wu et al., 2025b; Liu et al., 2025) or look-ahead evaluation (Yao et al., 2023; Wan et al., 2024; Park et al., 2025) to allocate rollout budgets across candidate prefixes: expanding selected prefixes while pruning low-value branches. However, this prefix reuse is mostly within-branch: parallel branches rarely share trajectory insights (e.g., verified facts or failure causes), leaving experience from pruned paths unused and surviving branches to explore in isolation. Ultimately, the search process remains largely memoryless at the systemic level, causing the model to repeatedly re-derive the same intermediate steps and revisit similar dead ends across extensive rollouts, leading to substantial redundant computation.

---
[1]Anonymous Institution, Anonymous City, Anonymous Region, Anonymous Country. Correspondence to: Anonymous Author <anon.email@domain.com>.

Preliminary work. Under review by the International Conference on Machine Learning (ICML). Do not distribute.

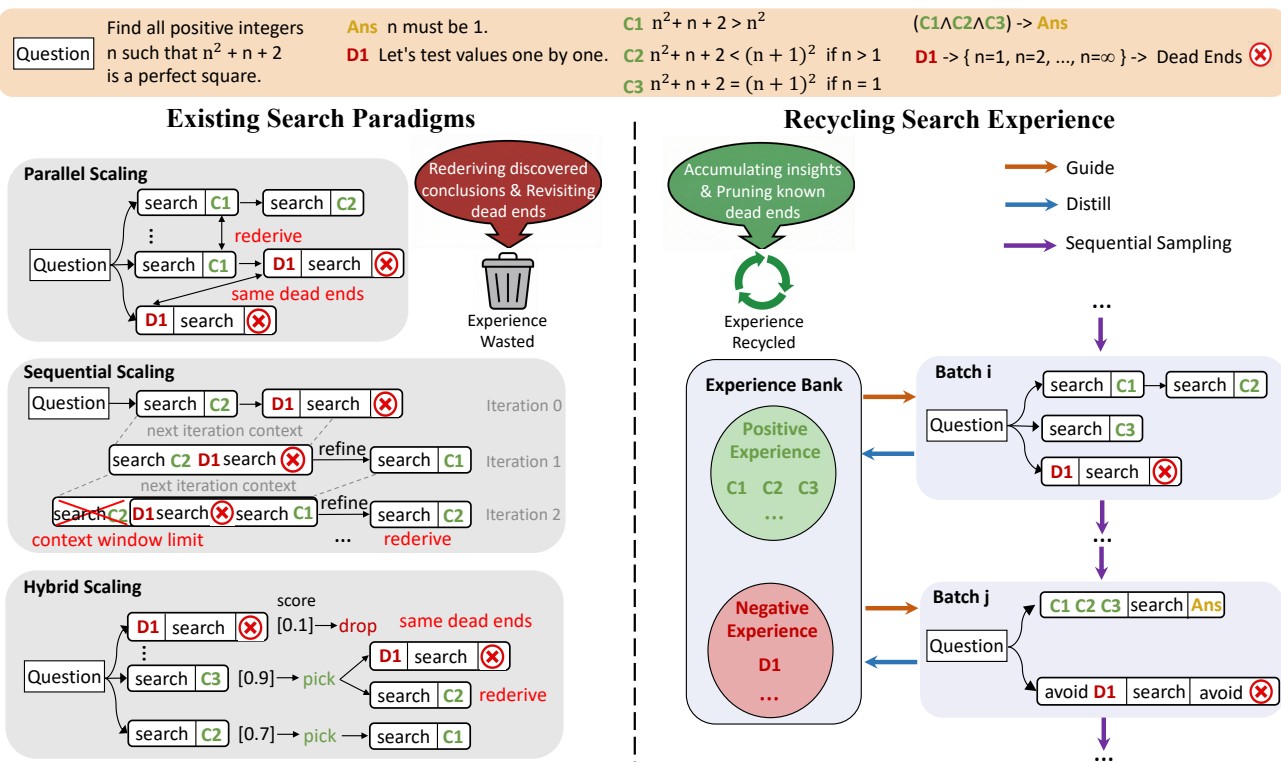

**Figure 1. From memoryless rollouts to experience-guided search. (Left)** Existing test-time scaling paradigms (parallel, sequential, and hybrid) largely treat rollouts as disposable: intermediate conclusions are repeatedly re-derived and dead ends are revisited across rollouts. **(Right)** Recycling Search Experience (RSE) runs rollouts in batches, distills reusable trajectory information into a shared Experience Bank, and conditions subsequent exploration on it. Positive recycling shares **intermediate conclusions** (e.g., $C_1$–$C_3$) to shortcut redundant derivations, while negative recycling records **failure patterns** (e.g., $D_1$) to prune known dead ends.

To address this inefficiency, we propose **Recycling Search Experience (RSE)**, a self-guided, training-free strategy that turns rollouts from disposable samples into reusable experience to actively guide the search process. Instead of viewing rollouts as isolated trials, we treat search as a cumulative process, where valuable insights from prior trajectories are fed back to guide later exploration. Leveraging the model's self-assessment capability (Weng et al., 2023; Dhuliawala et al., 2024), RSE distills valuable insights from rollouts into a shared experience bank without relying on external supervision, and explicitly conditions subsequent exploration on this bank. This mechanism enables two forms of experience reuse: *positive recycling*, where intermediate conclusions are shared to shortcut redundant derivations, and *negative recycling*, where encountered dead ends serve as constraints to prune the search space. By iterating this distill-and-guide loop, RSE turns test-time scaling from isolated trials into a cumulative, experience-guided search that concentrates compute on promising, unexplored regions of the solution space.

Intuitively, solving complex reasoning problems via independent sampling requires a single rollout to sequentially derive all necessary intermediate conclusions—a probabil-

ity that decays exponentially with problem complexity. In contrast, RSE effectively "checkpoints" valid intermediate conclusions, allowing the model to piece together the solution from fragmented partial successes rather than starting from scratch. We provide a theoretical analysis that formalizes and validates this intuition (see Appendix A).

To validate the effectiveness of RSE, we evaluate it on challenging mathematical reasoning benchmarks, including HMMT24, HMMT25 (Balunović et al., 2025), IMO-Bench (Luong et al., 2025), and HLE (Phan et al., 2025), across a broad range of rollout budgets and policy models. The empirical results show that RSE consistently outperforms strong baseline strategies under comparable computational budgets, highlighting its ability to enhance the overall efficiency of TTS.

To summarize, our main contributions are:

- We propose Recycling Search Experience (RSE), a training-free inference strategy that turns test-time search into a cumulative, experience-guided process by recycling rollout-level experience.

- We provide a theoretical analysis that formalizes the ef-

ficiency gains of RSE, validating the advantages of experience recycling over independent sampling in solving complex reasoning tasks.

- We empirically validate RSE on challenging mathematical reasoning benchmarks (HMMT24, HMMT25, IMO-Bench, HLE), demonstrating consistent gains over strong baselines across diverse models and rollout budgets.

## 2. Related Work

**Test-Time Scaling.** The paradigm of scaling test-time compute has emerged as a critical avenue for enhancing reasoning capabilities (OpenAI, 2024; DeepSeek-AI et al., 2025). Early approaches explored two primary dimensions: scaling depth via sequential refinement (Wei et al., 2022; Madaan et al., 2023; Shinn et al., 2023), which iteratively improves a single chain of thought; and scaling width via ensemble sampling (Wang et al., 2023b), which leverages the diversity of independent rollouts to marginalize out errors. To add structure beyond depth/width scaling, prior work explores tree-based search such as Tree of Thoughts (Yao et al., 2023) and MCTS (Liu et al., 2023; Wan et al., 2024), which allocate compute via explicit lookahead and rely on external supervision (e.g., labels or outcome reward models) to estimate the value of intermediate states. More recently, Process Reward Models (PRMs) provide step-level feedback for pruning and budget allocation, reducing the reliance on expensive lookahead simulation at inference (Lightman et al., 2024; Snell et al., 2024; Beeching et al., 2024; Wu et al., 2025b). Nevertheless, these approaches still rely on external supervision, which can limit their applicability in general scenarios. In this work, we focus specifically on self-guided strategies that scale efficiently without requiring ground-truth labels or external evaluators.

Within this line of research, the most relevant concurrent study is PaCoRe (Hu et al., 2026), which similarly integrates search history into the context. PaCoRe concatenates final answers from prior rollouts for consistency calibration, while discarding the intermediate thinking content, which could be mined as reusable experience. In contrast, RSE actively mines reusable experience from the entire exploration trajectory to guide subsequent exploration.

**Agentic Memory.** Memory mechanisms empower agents to persist information beyond limited context windows (Packer et al., 2023; Park et al., 2023; Hu et al., 2025). In the reasoning domain, existing methods primarily utilize memory for cross-query transfer, accumulating experience across different tasks. Prominent approaches include building skill libraries for code generation (Wang et al., 2023a) or retrieving relevant trajectories from historical databases (Ouyang et al., 2025) to aid similar future queries.

Similarly, Batch-of-Thought (Yang et al., 2026) facilitates cross-instance learning within a batch of questions. Distinct from these approaches that seek analogies from external history or parallel instances, our work targets intra-query memory reuse. RSE specifically recycles the exploration experience generated within the current search process to guide subsequent exploration and avoid redundant errors.

**LLM Context Engineering.** Context engineering studies how to systematically construct, process, and manage the information payload provided to LLMs under a limited context window, beyond prompt wording alone (Mei et al., 2025). A common direction is to improve information density by selecting or compressing long inputs while preserving task-relevant semantics, e.g., LLMLingua (Jiang et al., 2023; Pan et al., 2024) and Selective Context (Li et al., 2023), as well as long-horizon context summarization for agent settings (Wu et al., 2025a; Kang et al., 2025). RSE can be viewed through the same lens in a search setting: instead of retaining verbose, low-signal rollout traces, RSE distills the search process into compact, structured experience and maintains it via deduplication, maximizing the utility of prior search experience within a fixed context budget.

## 3. Methodology

Recycling Search Experience (RSE) is motivated by a simple principle: test-time search should not discard the valuable information contained in rollouts. We therefore treat search as a cumulative process, where rollouts from earlier rounds are distilled into reusable experience and fed back to guide later exploration under the same compute budget. The overall workflow of RSE is summarized in Algorithm 1, comprising three coordinated components: Batched Experience-Guided Search, Self-Guided Distillation, and Semantic Experience Deduplication.

### 3.1. Batched Experience-Guided Search

Standard parallel sampling ensures breadth but suffers from *isolation*, where rollouts cannot share intermediate discoveries. Conversely, purely sequential refinement enables reuse but is hindered by limited parallelism and context limits, often forcing truncation or compression of earlier attempts and keeping exploration local around the current draft. A natural question is *whether we can share experience during a fully parallel rollout*. In practice, mid-trajectory synchronization is difficult to define and implement robustly: evaluating partial reasoning states is unreliable (e.g., distinguishing an actual error from a complex intermediate step) (Pandit et al., 2025), and injecting global updates changes the conditioning context, which can disrupt the current reasoning trajectory and increase output variance (Laban et al., 2025).

These issues make real-time experience sharing impractical

without specific architectural changes. RSE therefore adopts a **batched experience-guided** protocol that offers a simple and stable coordination interface. We partition the rollout budget into $R$ rounds; in each round $r$, the model generates a batch of $K_r$ trajectories in parallel. Across rounds, the system maintains a global **Experience Bank** $\mathcal{B}_{r-1}$ that summarizes reusable insights distilled from all previous rounds. Before launching round $r$, we serialize $\mathcal{B}_{r-1}$ into the prompt (see Appendix D for details), so all rollouts in the batch start from a synchronized state. This design preserves within-round diversity while enabling cross-round information reuse, thereby balancing information reuse with practical inference latency.

### 3.2. Self-Guided Experience Distillation

Simply concatenating all previous trajectories into the context window is impractical due to length constraints and the low signal-to-noise ratio of raw reasoning chains. We leverage the model's intrinsic capability to critique its own reasoning and identify valuable experience without external supervision, avoiding the need for ground-truth labels or external reward models, thereby making RSE applicable to general search environments.

Based on this, we propose **Self-Guided Experience Distillation**. Instead of retaining entire trajectories, we employ a lightweight prompting step to distill each rollout $\omega$ into discrete, structured experience items (see Appendix D for details). Specifically, for every generated trajectory, the model extracts: (1) **Positive Experience ($\mathcal{E}^{pos}$):** Verified propositions or lemmas that serve as "truth anchors" for future batches; (2) **Negative Experience ($\mathcal{E}^{neg}$):** Critical pitfalls or strategic dead ends that act as "negative constraints." This explicit extraction transforms unstructured exploration logs into compact, actionable guidance, facilitating high-quality experience reuse for future exploration.

### 3.3. Semantic Experience Deduplication

Even with distilled items, the accumulated experience can rapidly grow over multiple rounds, risking context overflow. Moreover, parallel rollouts within a batch often exhibit high redundancy: simple steps or common errors tend to be discovered repeatedly by multiple trajectories. Without mitigation, these repetitive items could dominate the prompt, crowding out rarer, high-value insights (Table 4).

To address this, we employ a **Semantic Experience Deduplication** strategy to maintain the Experience Bank as a diverse set rather than a simple list. As detailed in Algorithm 1, we employ an incremental greedy selection strategy. By evaluating candidates against the dynamically updated Experience Bank, we ensure that each new entry is semantically distinct from all previously admitted items, effectively filtering out redundancy both from historical rounds and

---

**Algorithm 1** Recycling Search Experience (RSE)

---

**Require:** Problem $x$, model $\pi$, rounds $R$, batch sizes $\{K_r\}_{r=1}^R$, similarity threshold $\tau$.
**Ensure:** Final batch of trajectories $\Omega_R$.
 1: Initialize Experience Bank: $\mathcal{E}_0^{pos} \leftarrow \emptyset, \mathcal{E}_0^{neg} \leftarrow \emptyset$.
 2: **for** $r = 1$ **to** $R$ **do**
 3:    // Step 1: Batched Experience-Guided Search (Sec. 3.1)
 4:    $u_r \leftarrow \mathsf{Prompt}(x, (\mathcal{E}_{r-1}^{pos}, \mathcal{E}_{r-1}^{neg}))$.
 5:    Sample in parallel: $\Omega_r \leftarrow \{\omega_r^{(i)} \sim \pi(\cdot \mid u_r)\}_{i=1}^{K_r}$.
 6:    // Step 2: Self-Guided Experience Distillation (Sec. 3.2)
 7:    Initialize batch experience: $\Delta_r^{pos} \leftarrow \emptyset, \Delta_r^{neg} \leftarrow \emptyset$.
 8:    **for each** rollout $\omega \in \Omega_r$ **do**
 9:      $(\delta^{pos}, \delta^{neg}) \leftarrow \mathsf{Distill}(x, \omega)$
10:      $\Delta_r^{pos} \leftarrow \Delta_r^{pos} \cup \delta^{pos}; \quad \Delta_r^{neg} \leftarrow \Delta_r^{neg} \cup \delta^{neg}$.
11:    **end for**
12:    // Step 3: Semantic Experience Deduplication (Sec. 3.3)
13:    **for** type $\in \{pos, neg\}$ **do**
14:      $\mathcal{E}_r^{\text{type}} \leftarrow \mathcal{E}_{r-1}^{\text{type}}$
15:      **for each** $\delta \in \Delta_r^{\text{type}}$ **do**
16:        **if** $\max_{e \in \mathcal{E}_r^{\text{type}}} \mathsf{Sim}(\delta, e) < \tau$ **then**
17:          $\mathcal{E}_r^{\text{type}} \leftarrow \mathcal{E}_r^{\text{type}} \cup \{\delta\}$
18:        **end if**
19:      **end for**
20:    **end for**
21: **end for**
22: **return** $\Omega_R$.

---

within the current batch. This mechanism prevents context explosion by filtering out repetitive experiences, thereby maintaining high information density within the limited context window.

## 4. Experiments

In this section, we present the experimental evaluation of RSE. We begin by outlining the experimental setup, followed by the main results comparing RSE against baselines, and an in-depth analysis focusing on scalability, efficiency, reasoning dynamic and the impact of experience context construction strategies. Additionally, a concrete case analysis is provided in Appendix C.6.

### 4.1. Experimental Setup

**Benchmarks and Models.** We evaluate the proposed RSE strategy on four challenging benchmarks: HMMT24, HMMT25, IMO-AnswerBench, and a 100-sample math subset from Humanity's Last Exam (HLE-Math-text). These datasets serve as reliable proxies for advanced problem-solving capabilities requiring complex multi-step reasoning. We evaluate the proposed RSE strategy primarily on models specialized for complex reasoning tasks, spanning diverse

*Table 1.* Performance comparison across different models and iterations on four benchmarks (HMMT24, HMMT25, IMO-AnswerBench, HLE-Math-text). Values are reported as Pass@1 (%). "It0" denotes the base/initial performance, while "It1-3" represent subsequent iterative refinements. Gray values in It0 indicate the Base performance carried over for comparison. Best performance in each iteration is **bolded**.

| Method | HMMT24 | | | | HMMT25 | | | | IMO-AnswerBench | | | | HLE-Math-text | | | |
|---|---|---|---|---|---|---|---|---|---|---|---|---|---|---|---|---|
| | It0 | It1 | It2 | It3 | It0 | It1 | It2 | It3 | It0 | It1 | It2 | It3 | It0 | It1 | It2 | It3 |
| *Qwen3-30B-A3B-Thinking-2507* | | | | | | | | | | | | | | | | |
| Base | 57.4 | – | – | – | 69.0 | – | – | – | 50.5 | – | – | – | 24.0 | – | – | – |
| MV@128 | 60.9 | – | – | – | 69.8 | – | – | – | 54.1 | – | – | – | 28.6 | – | – | – |
| Self-Ref | 57.4 | 62.7 | 64.5 | 65.6 | 69.0 | 71.4 | 72.5 | 72.6 | 50.5 | 49.5 | 49.9 | 50.1 | 24.0 | 26.1 | 26.0 | 26.2 |
| PaCoRe | 57.4 | 70.6 | 72.2 | 73.1 | 69.0 | 78.4 | 79.7 | 80.2 | 50.5 | 57.3 | 57.8 | 58.0 | 24.0 | 38.0 | 40.2 | 40.7 |
| RSE | 57.4 | **71.9** | **73.7** | **74.4** | 69.0 | **81.3** | **82.9** | **83.9** | 50.5 | **59.2** | **60.1** | **60.3** | 24.0 | **39.9** | **43.0** | **44.8** |
| *Qwen3-4B-Thinking-2507* | | | | | | | | | | | | | | | | |
| Base | 42.6 | – | – | – | 54.0 | – | – | – | 42.1 | – | – | – | 14.2 | – | – | – |
| MV@128 | 43.3 | – | – | – | 56.7 | – | – | – | 46.4 | – | – | – | 14.7 | – | – | – |
| Self-Ref | 42.6 | 47.0 | 48.8 | 49.7 | 54.0 | 58.2 | 58.6 | 59.4 | 42.1 | 43.5 | 46.2 | 46.3 | 14.2 | 14.8 | 14.8 | 15.0 |
| PaCoRe | 42.6 | **56.7** | 54.6 | 54.9 | 54.0 | **69.6** | 70.5 | 70.0 | 42.1 | 46.0 | 48.2 | 49.0 | 14.2 | **19.6** | 19.4 | 19.5 |
| RSE | 42.6 | 54.3 | **55.1** | **56.0** | 54.0 | 68.6 | **72.2** | **73.5** | 42.1 | **48.6** | **49.6** | **49.6** | 14.2 | 19.5 | **20.2** | **20.8** |
| *Phi-4-Reasoning* | | | | | | | | | | | | | | | | |
| Base | 40.3 | – | – | – | 43.9 | – | – | – | 34.5 | – | – | – | 8.5 | – | – | – |
| MV@128 | 42.2 | – | – | – | 52.2 | – | – | – | 43.9 | – | – | – | 8.8 | – | – | – |
| Self-Ref | 40.3 | 41.4 | 42.3 | 43.0 | 43.9 | 46.8 | 48.5 | 49.9 | 34.5 | 35.5 | 35.1 | 34.7 | 8.5 | 8.1 | 8.0 | 8.0 |
| PaCoRe | 40.3 | 49.0 | 52.4 | 52.3 | 43.9 | 59.0 | 62.0 | 63.8 | 34.5 | 39.2 | 40.1 | 40.6 | 8.5 | 8.1 | 9.3 | 9.6 |
| RSE | 40.3 | **51.0** | **55.9** | **56.5** | 43.9 | **60.2** | **66.0** | **67.5** | 34.5 | **40.5** | **40.3** | **42.3** | 8.5 | **9.0** | **10.9** | **11.4** |

scales and architectures: QWEN3-30B-A3B-THINKING-2507, QWEN3-4B-THINKING-2507 (Yang et al., 2025), PHI-4-REASONING (Abdin et al., 2025), and DEEPSEEK-V3.2 (DeepSeek-AI, 2025). To further demonstrate the universality of RSE across training paradigms, we extend our analysis to general-purpose instruction-tuned models, specifically including QWEN3-30B-A3B-INSTRUCT-2507 and QWEN3-4B-INSTRUCT-2507.

**Baselines.** We compare RSE against three distinct categories of inference strategies: (1) **Standard Sampling**, evaluating the model's intrinsic performance; (2) **Majority Voting** (Wang et al., 2023b), mitigating stochasticity by aggregating consensus across multiple reasoning paths; (3) **Self-Refine** (Madaan et al., 2023), performing iterative refinement on a single reasoning trajectory; and (4) **PaCoRe** (Hu et al., 2026), embedding historical information via direct concatenation of past outputs.

**Evaluation Protocol.** Unless otherwise specified, we adhere to the following configurations to ensure rigorous comparison. For Standard Sampling, we conduct a single large-scale experiment consisting of $1,024$ independent stochastic rollouts to estimate the intrinsic pass@1 accuracy. For Majority Voting, results are derived from a budget of 128 rollouts. For both RSE and Sequential Optimization Baselines, we standardize the process into a common multi-round search framework. By default, this consists of 1 reference

initialization iteration 3 subsequent optimization iterations. Specifically, we maintain 8 parallel groups, each initialized with a population of 32 distinct reference responses. In each iteration, the method generates 32 new rollouts per group to serve as reference responses for the next round. To monitor the optimization progress, we report the pass@1 accuracy of the generated rollouts at each specific iteration. To ensure statistical reliability, all experiments are repeated three times except for Standard Sampling. Further implementation details are provided in Appendix B. The full set of prompt templates is detailed in Appendix D.

**Model-Specific Configurations.** We evaluate DEEPSEEK-V3.2 on the 100-sample HLE-Math-text subset exclusively. For PHI-4-REASONING, we implement specific context truncation to accommodate PaCoRe within the model's 32k window limit. More details are provided in Appendix B.2.

### 4.2. Main Results

The main experimental results across four benchmarks and three model architectures are summarized in Table 1. We observe consistent trends demonstrating the superiority of RSE over both standard sampling and sequential optimization baselines.

**RSE Achieves State-of-the-Art Performance.** As shown in Table 1, RSE consistently outperforms all baselines. Crucially, this superiority demonstrates robustness across both

model scales and training paradigms. On smaller models like QWEN3-4B-THINKING, RSE unlocks capabilities that were previously inaccessible, enabling it to reach 73.5% on HMMT25, competitively matching the baseline performance of the much larger 30B model (69.0%). This indicates that effective test-time search can compensate for parametric limitations to a significant extent. Furthermore, extended evaluations on standard instruction-tuned models (Table 2) confirm that RSE generalizes beyond reasoning-specialized architectures.

*Table 2.* Results of instruction-tuned models on HMMT24 and HMMT25.

| Method | HMMT24 | | | | HMMT25 | | | |
|---|---|---|---|---|---|---|---|---|
| | It0 | It1 | It2 | It3 | It0 | It1 | It2 | It3 |
| *Qwen3-30B-Instruct* | | | | | | | | |
| Base | 26.8 | – | – | – | 33.2 | – | – | – |
| MV@128 | 23.7 | – | – | – | 33.4 | – | – | – |
| Self-Ref | 26.8 | 39.5 | 41.6 | 43.7 | 33.2 | 45.2 | 48.3 | 50.7 |
| PaCoRe | 26.8 | **52.8** | 53.6 | 53.7 | 33.2 | 55.7 | 57.0 | 57.2 |
| RSE | 26.8 | 50.3 | **55.4** | **54.4** | 33.2 | **58.5** | **62.3** | **64.4** |
| *Qwen3-4B-Instruct* | | | | | | | | |
| Base | 21.7 | – | – | – | 27.1 | – | – | – |
| MV@128 | 21.8 | – | – | – | 26.7 | – | – | – |
| Self-Ref | 21.7 | 26.8 | 28.6 | 29.4 | 27.1 | 31.9 | 33.2 | 34.0 |
| PaCoRe | 21.7 | 35.0 | 38.2 | 40.0 | 27.1 | 43.2 | 44.4 | 44.8 |
| RSE | 21.7 | **37.8** | **40.4** | **41.5** | 27.1 | **44.1** | **47.0** | **49.4** |

**Scaling Dynamics: Higher Ceilings and Delayed Convergence.** Beyond mere cumulative gains, RSE exhibits a fundamentally superior scaling trajectory compared to baselines. First, RSE strictly dominates sequential baselines at every corresponding iteration, achieving higher pass@1 accuracy under identical interaction. Second, RSE raises the upper bound of performance. While methods like Self-Refine and PaCoRe typically exhibit early saturation, RSE continues to extract significant gains through Iteration 3, effectively delaying convergence to reach a higher asymptotic limit.

**Efficacy on Hard Reasoning Tasks.** On the more challenging benchmarks like HLE-Math-text, where "guessing" strategies like Majority Voting yield marginal improvements, RSE reliably extracts valid reasoning paths. This divergence highlights a critical distinction: while standard ensemble methods fail when the correct solution is not the dominant mode, RSE effectively navigates the search space to discover valid solutions that are initially assigned low probability, thereby overcoming the systematic errors of the base model.

**Generalization to Frontier Model.** We additionally extend our evaluation to DEEPSEEK-V3.2 on the HLE-Math-

text benchmark, as detailed in Table 3. Despite the high intrinsic baseline, RSE achieves a substantial gain of 13.5% at Iteration 2. These results align well with the trends observed in Table 1, further corroborating the robustness of our approach across varying model scales and capabilities.

*Table 3.* Performance of DEEPSEEK-V3.2 on the HLE-Math-text.

| Method | It0 | It1 | It2 |
|---|---|---|---|
| *Deepseek-V3.2 on HLE-Math-text* | | | |
| Base | 49.3 | – | – |
| MV@128 | 58.4 | – | – |
| PaCoRe | 49.3 | 58.1 | 57.9 |
| RSE | 49.3 | **59.2** | **62.8** |

### 4.3. Analysis

We analyze the effectiveness of RSE from three key perspectives: (1) Scalability and Efficiency, assessing performance gains across search depth, width, and compute cost; (2) Reasoning Dynamics, investigating how experience recycling influences exploration behavior; and (3) Experience Composition, dissecting the impact of distinct experience types and selection strategies. Additionally, we provide supplementary analysis in Appendix C, covering lexical analysis of reasoning traces, output truncation statistics, quality analysis of distilled experiences, and case analysis on an experience-guided reasoning case example. Unless otherwise specified, all experiments in analysis experiments are conducted using QWEN3-30B-A3B-THINKING-2507.

#### 4.3.1. SCALABILITY AND EFFICIENCY

We systematically dissect the scalability of RSE across three fundamental dimensions: search depth, search width, and compute flops. The results are summarized in Figure 2.

**Scaling Search Depth.** Figure 2(a) illustrates a distinct divergence in scaling behaviors (Majority Voting is excluded from this comparison, as its non-iterative nature lacks the mechanism to refine prior rollouts). PaCoRe hits a premature ceiling by Iteration 3, driven by the verification-centric bottleneck where repeated validation against fixed references yields rapid diminishing returns (detailed in Appendix C.1). In contrast, RSE sustains significant gains through Iteration 6, effectively raising the performance upper bound. Crucially, although Self-Refine also benefits from increased depth without immediate saturation, it exhibits a severe efficiency deficit, consistently trailing RSE by a significant margin in absolute pass@1. This indicates that RSE optimizes the search trajectory more effectively: by synthesizing diverse population-level experiences rather than iterating on isolated traces or checking against a static consensus, RSE significantly enhances the marginal utility of each search step.

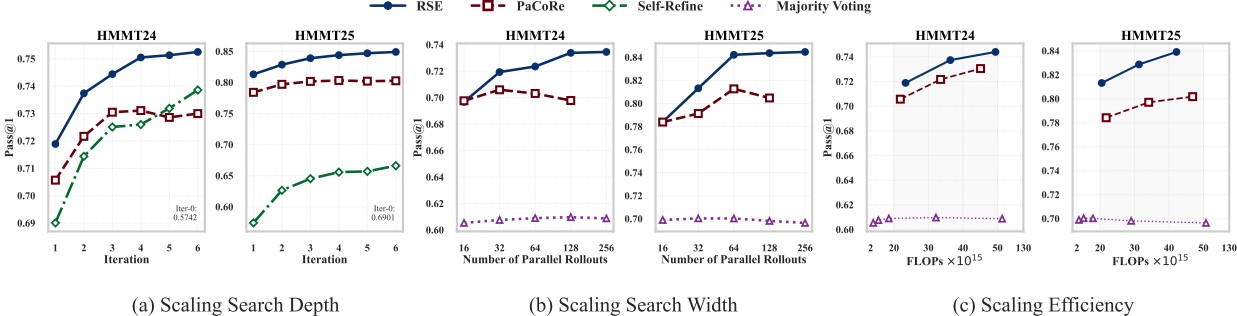

(a) Scaling Search Depth       (b) Scaling Search Width       (c) Scaling Efficiency

*Figure 2.* **Scalability and Efficiency Analysis of Test-Time Search.** We evaluate the scaling behaviors of different search strategies across three dimensions: (a) search depth, (b) search width, and (c) computational efficiency .

*Table 4.* Impact of the de-duplication threshold $\tau$ on experience retention and performance. We compare the number of retained positive experiences ($N_{pos}$) and negative experiences ($N_{neg}$) across iterations on HMMT24 and HMMT25. The best Pass@1 scores for each iteration are bolded.

| $\tau$ | Iteration 1 | | | Iteration 2 | | | Iteration 3 | | |
|---|---|---|---|---|---|---|---|---|---|
| | $N_{pos}$ | $N_{neg}$ | Pass@1 | $N_{pos}$ | $N_{neg}$ | Pass@1 | $N_{pos}$ | $N_{neg}$ | Pass@1 |
| **HMMT24** | | | | | | | | | |
| 0.6 | 19.4 | 14.9 | 70.8 | 19.4 | 15.5 | **74.1** | 19.2 | 14.6 | 74.1 |
| 0.7 | 36.6 | 30.1 | 71.1 | 40.4 | 34.3 | 73.3 | 41.8 | 34.6 | 74.2 |
| 0.8 | 66.1 | 55.8 | **71.9** | 85.2 | 72.5 | 73.7 | 95.3 | 80.9 | **74.4** |
| 0.9 | 113.5 | 89.2 | 70.8 | 177.7 | 123.2 | 72.4 | 225.8 | 153.2 | 72.9 |
| 1.0 | 161.7 | 94.5 | 70.3 | 339.8 | 149.9 | 72.4 | 526.4 | 202.2 | 73.3 |
| **HMMT25** | | | | | | | | | |
| 0.6 | 18.4 | 15.5 | 80.2 | 20.7 | 15.4 | 82.9 | 20.9 | 14.9 | 83.0 |
| 0.7 | 34.6 | 30.6 | 81.5 | 42.7 | 34.9 | 83.4 | 45.0 | 36.5 | 84.4 |
| 0.8 | 62.7 | 55.1 | 81.3 | 89.5 | 74.6 | 82.9 | 101.4 | 85.1 | 83.9 |
| 0.9 | 107.7 | 80.4 | **82.5** | 185.7 | 135.7 | **84.4** | 237.7 | 173.1 | **85.1** |
| 1.0 | 158.4 | 89.8 | **82.5** | 330.9 | 165.1 | 84.1 | 505.1 | 230.0 | 84.4 |

**Scaling Search Width.** Figure 2(b) investigates the impact of scaling the reference number ($N_{ref}$). The results underscore a critical divergence in scaling behaviors. Majority Voting exhibits a flat performance trajectory, indicating that simply increasing rollout width yields negligible marginal utility without a mechanism to aggregate inter-path experiences. While PaCoRe attempts to utilize such experience via concatenation, it faces severe scalability bottlenecks. First, PaCoRe is fundamentally constrained by the model's context window. This limitation imposes a hard ceiling on usable references, compelling us to omit the evaluation at $N_{ref} = 256$ as the concatenated input exceeds the 256k context window of QWEN3-30B-A3B-THINKING-2507. Furthermore, we observe that performance actively regresses beyond a certain threshold. We hypothesize that this stems from attention dispersion arising from the excessively long context (Liu et al., 2024). RSE circumvents these limitations by decoupling experience accumulation from context length. It effectively assimilates heterogeneous insights from a growing reference pool without inducing

context saturation, thereby maintaining robust performance gains and continuous reasoning refinement as width scales.

**Scaling Efficiency.** Finally, to rigorously assess the cost-effectiveness of our method, we benchmark performance against the total computational cost (measured in FLOPs; detailed calculation protocols are provided in Appendix B.4). As shown in Figure 2(c), RSE establishes a superior Pareto frontier compared to both PaCoRe and Majority Voting. Notably, MV@128 exhibits a nearly flat trajectory, indicating that brute-force sampling yields minimal returns beyond a certain scale. In contrast, RSE demonstrates a steep ascent, delivering the highest accuracy gains per unit of compute budget. This confirms that structured experience recycling represents a significantly more compute-optimal strategy for scaling test-time inference compared to simple aggregation or verification-based concatenation.

### 4.3.2. REASONING DYNAMICS ON HARD PROBLEMS.

Figure 5 reveals that the issue of output truncation is effectively mitigated from the first iteration onwards. To decouple genuine reasoning improvements from mere format-level restoration (i.e., the resolution of truncation errors), we employ a specialized metric for this analysis: *Non-Truncated Pass@1*. Calculated strictly on completed rollouts, this metric isolates the model's logical derivation capabilities. Leveraging this metric, we conduct an aggregated analysis across four benchmarks, specifically filtering for "Hard Problems", defined as instances where the baseline (i.e., QWEN3-30B-A3B-THINKING-2507 Standard Sampling) *Non-Truncated Pass@1* falls within the $[0, 0.5]$ interval (Snell et al., 2024).

Figure 3 highlights RSE's superior robustness in hard samples, and this advantage is particularly pronounced in the extremely-hard bracket ($[0, 0.1]$) where valid solutions are scarce. In the extremely-hard bracket, while PaCoRe stagnates due to a behavioral collapse towards passive reference verification, RSE sustains performance gains by preserving

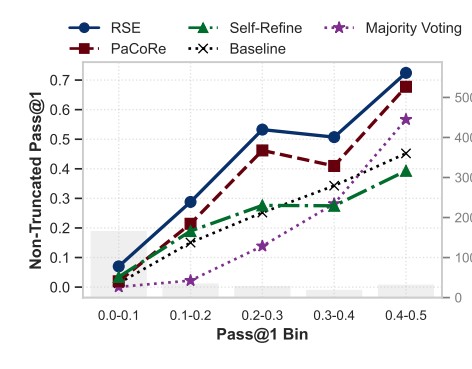

*Figure 3.* Non-truncated Pass@1 across varying difficulty problems. Samples are stratified by baseline Non-truncated Pass@1, with gray bars indicating the sample count distribution per bin.

independent exploration capacity. Analysis of reasoning dynamics confirms that RSE achieves superior efficiency by pruning redundancy, as evidenced by the sharp reduction in computational overhead from 23k to 8.4k tokens in Table 8. Crucially, despite this condensation, lexical analysis in Figure 4 demonstrates the persistence of cognitive markers like "*wait*", indicating that RSE leverages experience to guide the search while keeping the deductive chain intact. Unlike PaCoRe which abandons deep cognition in favor of checking against a potentially flawed consensus, RSE preserves active deduction, enabling it to discover valid solutions even when initial reference pool lacks high-quality answers.

### 4.3.3. ANALYSIS OF EXPERIENCE COMPOSITION AND UTILIZATION

We systematically investigate the impact of context construction across three dimensions: the granularity of experience filtering (deduplication threshold), the mechanism of information retention (diversity vs. frequency), and the synergy of specific experience components (positive vs. negative).

**Sensitivity to Deduplication Threshold.** Table 4 exhibits a non-monotonic performance trajectory peaking around $\tau = 0.8$. This trend highlights a clear trade-off: lower thresholds ($\tau < 0.7$) cause information loss by over-merging distinct paths, while higher thresholds ($\tau \to 1.0$) introduce redundancy-induced bias. Thus, a moderate threshold is essential to optimally balance reasoning diversity and context efficiency.

**Distinctness vs. Consistency.** To isolate the impact of information distinctness versus consistency reinforcement, we aligned the experience budget $N$ with the yield of our distinctness-driven selection. The baseline employs random sampling from the raw pool to explicitly introduce consistency information (i.e., high-frequency experiences). The sustained superiority of the distinctness-driven approach (Table 10) confirms that information richness facilitates rea-

*Table 5.* **Component-wise Ablation Study.** We evaluate the impact of different experience types on search performance. "Positive Only" and "Negative Only" denote using exclusively positive experience or negative experience, respectively. Performance is reported as Pass@1 (%).

| Configuration | HMMT-24 | | | HMMT-25 | | |
|---|---|---|---|---|---|---|
| | Iter 1 | Iter 2 | Iter 3 | Iter 1 | Iter 2 | Iter 3 |
| Baseline (No Exp.) | 57.4 | 57.4 | 57.4 | 69.0 | 69.0 | 69.0 |
| + Positive Exp. Only | 71.5 | 72.9 | 73.2 | 79.8 | 81.6 | 83.3 |
| + Negative Exp. Only | 71.8 | 73.2 | 73.9 | 79.7 | 81.0 | 81.6 |
| **+ Both (Full RSE)** | **71.9** | **73.7** | **74.4** | **81.3** | **82.9** | **83.9** |

soning more effectively than pure consistency, as diverse contexts provide broader logical references to accelerate solution derivation.

**Synergy of Experience Components.** Finally, we isolate the contributions of Positive Experiences and Negative Experiences to verify their individual necessity and combined synergy. The results, detailed in Table 5, reveal three key insights. First, both components independently yield substantial improvements over the baseline. Second, the two types of experience exhibit strong complementarity. Combining them (Full RSE) consistently outperforms using either in isolation across all iterations and benchmarks. This suggests that Positive Experiences and Negative Experiences address distinct reasoning failures: the former guides the model towards verified paths, while the latter explicitly blocks known dead ends, effectively pruning the search space from both directions.

## 5. Conclusions

In this work, we identify and address the systemic inefficiency of current "memoryless" test-time search, where valuable intermediate insights are largely discarded after every rollout. To bridge this gap, we introduce Recycling Search Experience (RSE), a self-guided, training-free strategy that turns test-time search from a series of isolated trials into a cumulative process. By actively distilling raw trajectories into structured intermediate conclusions and negative constraints, RSE enables models to prune explored dead ends and accelerate valid derivations without external supervision. Theoretically, we provide an analysis that formalizes the efficiency gains of RSE, validating its advantage over independent sampling in solving complex reasoning tasks. Empirically, extensive evaluations on challenging mathematical benchmarks demonstrate that RSE consistently establishes a superior Pareto frontier compared to strong baselines. Ultimately, our findings suggest a critical paradigm shift: maximizing the potential of test-time compute requires not merely increasing the volume of rollouts, but optimizing the quality of exploration by transforming disposable trials into cumulative experience.

## Impact Statement

This paper presents work whose goal is to advance the field of Machine Learning. There are many potential societal consequences of our work, none which we feel must be specifically highlighted here.

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

# A. Theory: Recycling Search Experience Improves the Probability of Finding a Correct Solution

## A.1. Setup: required conclusions and a verified-rollout oracle

**Definition A.1** (Required intermediate conclusions (analysis device)). Fix a problem instance $x$ and a prompting template. Let $\mathcal{C} = \{c_1, \ldots, c_L\}$ be a finite set of *required intermediate conclusions* such that a fully correct solution can be produced whenever all elements of $\mathcal{C}$ are available.[1] A rollout output is considered *correct* if its verified content contains all required conclusions:

$$\mathrm{Correct}(R) := \mathbf{1}\{\mathcal{C} \subseteq R\}.$$

**Definition A.2** (Verified rollout oracle). Let $\mathcal{M} \subseteq 2^{\mathcal{C}}$ be a family of *experience states* that contains $\varnothing$ and is closed under union. A single model call (generation + perfect verifier) is abstracted as a stochastic set-valued oracle

$$F : \mathcal{M} \times \Omega \to \mathcal{M}, \qquad R = F(S; \omega),$$

where $S \in \mathcal{M}$ is the current verified *experience* injected into the prompt, and $R \in \mathcal{M}$ is the set of conclusions verified correct in this rollout (including any injected ones).

## A.2. Two procedures and the same success notion

**Definition A.3** (Baseline vs. Recycling Search Experience (RSE)). Fix a rollout budget $N \in \mathbb{N}$ and randomness $\omega_1, \ldots, \omega_N$.

**Baseline (independent-from-scratch sampling).** For $t = 1, \ldots, N$, sample

$$B_t := F(\varnothing; \omega_t).$$

Define the probability of finding at least one correct solution within budget $N$ as

$$P_{\mathrm{succ}}^{\mathrm{base}}(N) := \mathbb{P}\Big(\exists t \leq N : \mathcal{C} \subseteq B_t\Big).$$

**Recycling Search Experience (RSE).** Initialize the Experience Bank $E_0 = \varnothing$. For $t = 1, \ldots, N$, sample

$$R_t := F(E_{t-1}; \omega_t), \qquad E_t := E_{t-1} \cup R_t.$$

Define

$$P_{\mathrm{succ}}^{\mathrm{rse}}(N) := \mathbb{P}\Big(\exists t \leq N : \mathcal{C} \subseteq R_t\Big).$$

Both procedures declare success under the same predicate: *"there exists a rollout whose verified output contains all required conclusions"*; the only difference is whether previously verified experience may be injected into later rollouts.

---

[1] This is an abstract analysis device: we do not assume $\mathcal{C}$ is observable. It captures the idea that solving the problem may require collecting multiple key facts or lemmas.

## A.3. Assumptions

**Assumption A.4** (Perfect verification and persistence of injected experience). For all $S \in \mathcal{M}$ and all $\omega \in \Omega$,

$$S \subseteq F(S; \omega).$$

That is, any previously verified experience injected into the prompt is guaranteed to be included in the current rollout's verified output.

**Assumption A.5** (No-harm under sound experience injection (monotonicity)). For any $S, S' \in \mathcal{M}$ with $S \subseteq S'$ and any $\omega \in \Omega$,

$$F(S; \omega) \subseteq F(S'; \omega).$$

## A.4. Main result: distribution-free dominance of success probability

**Theorem A.6** (RSE is no worse than independent sampling). *Under Assumptions A.4–A.5, for every budget $N \geq 1$,*

$$P_{\mathrm{succ}}^{\mathrm{rse}}(N) \geq P_{\mathrm{succ}}^{\mathrm{base}}(N).$$

*Moreover, this dominance holds without any independence assumption on $\omega_1, \ldots, \omega_N$.*

*Proof.* Couple the two procedures on the same randomness sequence $(\omega_t)_{t=1}^{N}$. For each $t$, since $E_{t-1} \supseteq \varnothing$, Assumption A.5 implies the pointwise inclusion

$$B_t = F(\varnothing; \omega_t) \subseteq F(E_{t-1}; \omega_t) = R_t.$$

Hence whenever $\mathcal{C} \subseteq B_t$ holds for some $t$, we also have $\mathcal{C} \subseteq R_t$ for the same $t$. Therefore,

$$\Big\{\exists t \leq N : \mathcal{C} \subseteq B_t\Big\} \subseteq \Big\{\exists t \leq N : \mathcal{C} \subseteq R_t\Big\}.$$

Taking probabilities yields $P_{\mathrm{succ}}^{\mathrm{rse}}(N) \geq P_{\mathrm{succ}}^{\mathrm{base}}(N)$. ∎

## A.5. A toy independent-coverage model: closed form and a sample-complexity gap

**Assumption A.7** (Additive experience model (toy)). Experience injection only guarantees the persistence of previously verified items and does not affect the distribution of *newly discovered* verified conclusions:

$$F(S; \omega) = S \cup F(\varnothing; \omega), \qquad \forall S \in \mathcal{M}, \ \omega \in \Omega.$$

**Corollary A.8** (Closed-form success probabilities under independent coverage). *Assume the from-scratch oracle follows an* independent coverage model*: for each $j \in [L]$ and each rollout $t$, the indicator $\mathbf{1}\{c_j \in F(\varnothing; \omega_t)\}$ is* Bernoulli$(p_j)$ with $p_j \in (0, 1)$, *independent across $t$ and $j$.*

*Under Assumption A.7,*

$$P_{\text{succ}}^{\text{base}}(N) = 1 - \Big(1 - \prod_{j=1}^{L} p_j\Big)^N,$$

$$P_{\text{succ}}^{\text{rse}}(N) = \prod_{j=1}^{L} \Big(1 - (1 - p_j)^N\Big), \tag{1}$$

*and in particular $P_{\text{succ}}^{\text{rse}}(N) \geq P_{\text{succ}}^{\text{base}}(N)$ for all $N$.*

*Moreover, letting $p_{\min} := \min_j p_j$ and target failure probability $\delta \in (0,1)$, it suffices to take*

$$N \geq \frac{1}{p_{\min}} \log \frac{L}{\delta} \quad \Rightarrow \quad P_{\text{succ}}^{\text{rse}}(N) \geq 1 - \delta,$$

*whereas a sufficient condition for $P_{\text{succ}}^{\text{base}}(N) \geq 1 - \delta$ is*

$$N \geq \frac{1}{\prod_{j=1}^{L} p_j} \log \frac{1}{\delta}.$$

*In the homogeneous case $p_j = p$, this exhibits an exponential gap in $L$: $N_{\text{base}} = \Omega\big(p^{-L} \log(1/\delta)\big)$ versus $N_{\text{rse}} = O\big(p^{-1} \log(L/\delta)\big)$.*

*Proof.* Under the independent coverage model, a single from-scratch rollout contains all conclusions with probability $q := \prod_{j=1}^{L} p_j$, hence $P_{\text{succ}}^{\text{base}}(N) = 1 - (1 - q)^N$.

Under Assumption A.7, the $t$-th RSE rollout satisfies

$$R_t = F(E_{t-1}; \omega_t) = E_{t-1} \cup F(\varnothing; \omega_t),$$

so $R_t$ becomes correct at the first time $t$ when the *last missing* conclusion in $\mathcal{C} \setminus E_{t-1}$ appears in $F(\varnothing; \omega_t)$. Equivalently, RSE succeeds within budget $N$ iff every $c_j$ appears at least once in the from-scratch components $\{F(\varnothing; \omega_t)\}_{t=1}^{N}$.

For each $j$, the probability that $c_j$ never appears in $N$ i.i.d. trials is $(1 - p_j)^N$, hence it appears at least once with probability $1 - (1 - p_j)^N$. Independence across $j$ yields $P_{\text{succ}}^{\text{rse}}(N) = \prod_{j=1}^{L} (1 - (1 - p_j)^N)$.

For the sample-complexity bound,

$$1 - P_{\text{succ}}^{\text{rse}}(N) = \mathbb{P}\Big(\exists j \in [L] : c_j \text{ never appears in } N \text{ trials}\Big)$$

$$\leq \sum_{j=1}^{L} (1 - p_j)^N$$

$$\leq L e^{-N p_{\min}}, \tag{2}$$

so choosing $N \geq \frac{1}{p_{\min}} \log \frac{L}{\delta}$ makes the failure probability at most $\delta$. The baseline bound follows similarly from $(1 - q)^N \leq \delta$ and $(1 - q)^N \leq e^{-Nq}$.

## B. Implementation Details

### B.1. Sampling Configuration

All experiments are executed in parallel on a cluster with 512 NVIDIA H800 GPUs, where each individual run is allocated to 8 GPUs (except 2 GPUs for Phi-4-Reasoning). To ensure reproducibility and fair comparison, we strictly adhered to the official recommended sampling hyperparameters for each model family:

- Qwen3-Thinking-Series: Temperature $T = 0.6$, top-$p = 0.95$, top-$k = 20$, with a maximum generation length of 38k tokens.

- Qwen3-Instruct-Series: Temperature $T = 0.7$, top-$p = 0.8$, top-$k = 20$, with a maximum generation length of 38k tokens.

- Phi-4-reasoning: Temperature $T = 0.8$, top-$p = 0.95$, top-$k = 50$, with a maximum generation length of 32k tokens.

- DeepSeek-V3.2: Temperature $T = 1.0$, top-$p = 0.95$, with a maximum generation length of 64k tokens. All DeepSeek evaluations were performed via the official API.[2]

**De-duplication Settings.** To maintain diversity within the retrieved context, we implemented a semantic de-duplication step. We utilized the `all-MiniLM-L6-v2`[3] model to encode response candidates and applied a cosine similarity threshold of $0.8$ to filter out redundant samples.

### B.2. Model-Specific Adaptations

**Evaluation Scope for Deepseek-V3.2.** For DEEPSEEK-V3.2, we omit evaluations on HMMT24 and HMMT25 given its reported near-saturation performance (90.2% pass@1 accuracy on HMMT25) (DeepSeek-AI, 2025). Additionally, constrained by computational costs, we exclude IMO-AnswerBench and restrict our assessment of DEEPSEEK-V3.2 exclusively to the most demanding HLE-Math-text 100-sample subset.

**Context Window Adaptation for Phi-4.** The limited context window of PHI-4-REASONING (32k) poses a distinct challenge for PaCoRe, which fundamentally relies on aggregating multiple historical reasoning traces. As detailed in Table 7, our analysis reveals an average reasoning length of $\approx$12.9k tokens), with the 95th percentile (P95) reaching 31.6k. Given that the prompt overhead must also be accommodated, these generation lengths effectively saturate

---

[2] https://platform.deepseek.com/usage
[3] https://huggingface.co/sentence-transformers/all-MiniLM-L6-v2

the model's entire 32k capacity. To guarantee sufficient headroom for the generation phase, we reserve a 12k token generation buffer, capping the total input prompt at 20k tokens. Leveraging the prior that reasoning paths tend to become more concise and convergent in subsequent iterations, we aggressively truncate individual reference responses to a maximum of 4k tokens. This adaptation highlights the critical dependency of context-augmented consistency strategies on the underlying model's context capacity. Additionally, as Phi-4 implies a fixed system prompt structure, we prepended system instructions to the user prompt.

*Table 6.* Token Length Statistics for Phi-4 Reasoning Content. The 95th percentile (P95) length ($\approx$31.6k) nearly exhausts the 32k context window, leaving negligible space for input prompts and necessitating the truncation strategy.

| Field | Average | Median | P95 |
|---|---|---|---|
| Reasoning Content | 12.9k | 10.7k | 31.6k |
| Final Text Answer | 1.1k | 1k | 2k |

### B.3. HLE Benchmark Subset Construction

We utilized the text-only math subset of the Humanity's Last Exam (HLE) benchmark. Due to the prohibitive computational cost of performing iterative reasoning evaluations on the full subset (976 samples), we constructed a representative subset using a difficulty-stratified sampling strategy. First, we assessed the difficulty of all samples using QWEN3-30B-A3B-THINKING-2507. We performed 128 standard sampling rollouts per question and calculated the Pass@1 score. To focus on problems with significant headroom for improvement, we divided the samples into 10 difficulty bins based on Pass@1, ranging from 0.0 to 0.5 with a step size of 0.05. From each bin, we randomly selected 10 samples, resulting in a balanced and tractable subset of 100 samples that preserves the difficulty distribution of the challenging regime.

### B.4. Computational Cost Estimation

To explicitly quantify the computational efficiency of different inference strategies, we estimate the Floating Point Operations (FLOPs) following the methodology used in the NVIDIA NeMo framework (nem, 2025).

**Token Consumption Dynamics.** We calculate the cumulative FLOPs by tracking the exact number of tokens processed at each search iteration $t$. Let $n$ be the reference width (number of parallel rollouts). Let $L_{\text{reason}}^t$ and $L_{\text{response}}^t$ denote the length of the reasoning content and final response generated at iteration $t$, respectively.

**PaCoRe** PaCoRe conditions each generation on the full set of $n$ references from the preceding iteration. For iteration

$t + 1$:

$$\mathcal{T}_{\text{in}}^{(t+1)} \approx n \cdot \left( n \cdot L_{\text{response}}^t \right) \tag{3}$$

$$\mathcal{T}_{\text{out}}^{(t+1)} = n \cdot \left( L_{\text{reason}}^{t+1} + L_{\text{response}}^{t+1} \right) \tag{4}$$

Here, the term $n \cdot n$ reflects that each of the $n$ new rollouts essentially processes the concatenation of $n$ historical references.

**RSE** RSE incorporates an intermediate *Experience Distillation* phase while maintaining linear input complexity for generation. Let $L_{\text{distill\_prompt}}$ denote the instruction length for experience extraction. Crucially, we define $L_{\text{context}}^{(t+1)}$ as the *comprehensive* input length for the reasoning phase, encompassing the system prompt, problem description, and the aggregated experience context constructed from previous iterations. Let $L_{\text{distill\_out}}^t$ represent the total output tokens generated during the distillation process (including both extraction reasoning and the formalized experience). The total token consumption for iteration $t + 1$ is calculated as:

$$\mathcal{T}_{\text{in}}^{(t+1)} = \underbrace{n \cdot (L_{\text{distill\_prompt}} + L_{\text{reason}}^t + L_{\text{response}}^t)}_{\text{Distillation Input}}$$

$$+ \underbrace{n \cdot L_{\text{context}}^{(t+1)}}_{\text{Reasoning Input}} \tag{5}$$

$$\mathcal{T}_{\text{out}}^{(t+1)} = \underbrace{n \cdot L_{\text{distill\_out}}^t}_{\text{Distillation Output}}$$

$$+ \underbrace{n \cdot (L_{\text{reason}}^{t+1} + L_{\text{response}}^{t+1})}_{\text{Reasoning Output}} \tag{6}$$

The detailed token statistics used for these calculations are provided in Table 7.

*Table 7.* **Token Statistics for FLOPs Estimation.** We detail the token counts for each phase. For PaCoRe, we separate Thinking ($L_{\text{think}}$) and Response ($L_{\text{resp}}$) lengths, as its input complexity scales quadratically with $L_{\text{resp}}$. For RSE, Iter-0 represents standard sampling (no distillation/context). Note the stark contrast in reasoning length ($L_{\text{think}}$ vs $L_{\text{gen}}$) starting from Iter-1.

| Dataset | Iter | PaCoRe | | RSE (Ours) | | |
|---|---|---|---|---|---|---|
| | | $L_{\text{think}}$ | $L_{\text{resp}}$ | $L_{\text{distill\_out}}$ | $L_{\text{context}}$ | $L_{\text{gen}}$ |
| HMMT-24 | 0 | 23,965 | 1,013 | – | – | 24,978 |
| | 1 | 2,157 | 926 | 2,638 | 8,137 | 13,760 |
| | 2 | 1,729 | 924 | 3,403 | 12,860 | 12,860 |
| | 3 | 1,522 | 941 | 3,734 | 12,440 | 12,303 |
| HMMT-25 | 0 | 22,264 | 1,038 | – | – | 23,302 |
| | 1 | 1,951 | 960 | 2,701 | 7,973 | 11,228 |
| | 2 | 2,017 | 994 | 3,611 | 10,313 | 10,313 |
| | 3 | 2,031 | 1,031 | 4,061 | 11,692 | 9,786 |

## C. Additional Analysis

### C.1. Lexical Analysis of Reasoning Dynamics

To investigate the cognitive mechanisms driving the performance divergence observed in Figure 2(a), we conducted a lexical analysis of the generated reasoning content. Figure 4 presents the word clouds derived from the reasoning traces of PaCoRe and RSE.

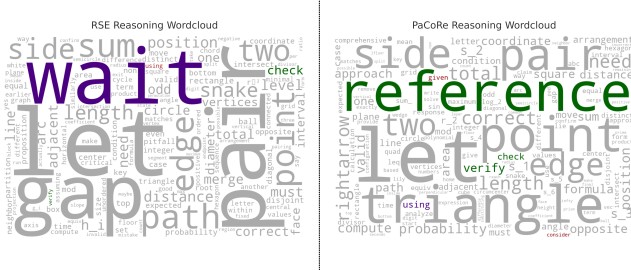

*Figure 4.* Word Cloud Analysis of Reasoning Content. Left: RSE; Right: PaCoRe.

**The Verification-Centric Bottleneck.**    The PaCoRe word cloud (Right) is dominated by meta-cognitive verification terms such as `"reference"`, `"verify"`, and `"check"`. This lexical distribution reveals a fundamental shift in the model's behavior: instead of engaging in independent problem-solving, the model repurposes its compute budget to validate its generation against the concatenated historical references. We term this the **Verification-Centric Bottleneck**. While this mechanism effectively filters out obvious inconsistencies in early iterations, it creates a closed feedback loop. Once the reference pool stabilizes (even on a suboptimal consensus), the model ceases to generate novel deductive steps, leading to the rapid diminishing returns and premature saturation observed in our scalability experiments.

**Preservation of Independent Derivation in RSE.**    In stark contrast, the RSE word cloud (Left) remains dominated by procedural reasoning markers such as `"wait"` and `"let"`. This lexical distribution confirms that RSE avoids collapsing into a shallow verification of reference content. Instead, it preserves the intrinsic reasoning patterns, ensuring that the generation process remains driven by active deduction rather than passive checking.

### C.2. Quantitative Evidence of Mode Collapse.

The qualitative shift identified in the lexical analysis (Appendix C.1) is further corroborated by quantitative metrics on reasoning length (Table 8) and answer entropy (Table 9). As shown in Table 8, PaCoRe exhibits a precipitous length collapse, with the average reasoning budget shrinking by over 90% (from 23k to 1.8k tokens) by Iteration 3. This

drastic reduction aligns with the Verification-Centric Bottleneck: verification is inherently less compute-intensive than generation. Once the model pivots to merely checking against references, it abandons the deep cognitive processes required for independent deduction. Concurrently, Table 9 reveals that PaCoRe's answer entropy decays to near-zero (0.0606), quantitatively confirming the occurrence of mode collapse. In stark contrast, RSE sustains a substantial reasoning length ($\approx$8.4k tokens) and higher entropy (0.3319), demonstrating that it preserves the capacity for active, diverse exploration throughout the iterative process.

*Table 8.* Average length of the generated `reasoning_content` across different iterations. RSE maintains a substantial reasoning budget, whereas PaCoRe exhibits a sharp length collapse.

| Method | Iter 0 | Iter 1 | Iter 2 | Iter 3 |
|---|---|---|---|---|
| PaCoRe | 23,309 | 3,935 | 2,184 | 1,885 |
| RSE (Ours) | | 10,696 | 9,044 | 8,417 |

*Table 9.* Answer entropy dynamics during search iteration.

| Method | Iter-0 | Iter-1 | Iter-2 | Iter-3 |
|---|---|---|---|---|
| RSE | 1.3264 | 0.5047 | 0.3779 | 0.3319 |
| PaCoRe | 1.3264 | 0.3603 | 0.1359 | 0.0606 |

### C.3. Impact of Context Composition.

Table 10 provides the detailed numerical breakdown for the ablation study regarding information distinctness versus consistency reinforcement.

*Table 10.* We compare context selection strategies under identical shot counts. *Dedup Sampling* uses the set of unique experiences identified by our method ($N$ unique items). *Random Sampling* selects an equal number ($N$) of examples from the raw pool. This setup isolates the effect of information diversity (Distinctness) versus frequency reinforcement (Consistency).

| Dataset | Iter | Random Sampling (Consistency-Biased) | Dedup Sampling (Distinctness-Driven) |
|---|---|---|---|
| HMMT24 | It-1 | 69.4 | **71.9** |
| | It-2 | 71.0 | **73.7** |
| | It-3 | 71.8 | **74.4** |
| HMMT25 | It-1 | 80.3 | **81.3** |
| | It-2 | 81.6 | **82.9** |
| | It-3 | 82.4 | **83.9** |

### C.4. Truncated Completions Analysis

Figure 5 illustrates the truncated completion rates varying across problem difficulty bins (grouped by pass@1 intervals).

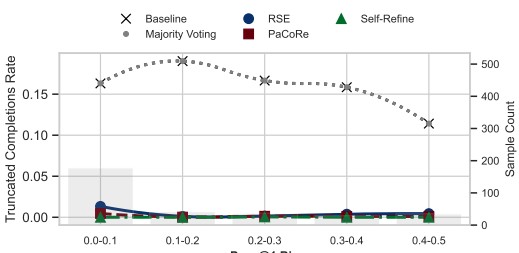

*Figure 5.* Truncated rate varying difficulty problems.

### C.5. Quality Verification of Distilled Experiences

To verify the reliability of our distilled experiences, we employed GEMINI-3-PRO-PREVIEW (DeepMind, 2025) as an automated validator. The model was prompted to evaluate each experience against the original problem, determining whether Positive Experiences are mathematically valid and whether Negative Experiences describe genuine logical flaws. The validation prompt is provided in Figure 11.

Table 11 reports the validation results. The distilled experiences achieve an overall accuracy of 81.06% on HMMT24 and 83.89% on HMMT25. These high validity scores confirm that our extraction module effectively distills correct and meaningful reasoning signals from the raw rollouts.

*Table 11.* Quality Verification of Distilled Experiences. Accuracy of extracted experiences as validated by GEMINI-3-PRO-PREVIEW.

| Dataset | Positive Exp. | Negative Exp. | Overall |
|---------|---------------|---------------|---------|
| HMMT24  | 84.48         | 77.61         | 81.06   |
| HMMT25  | 86.25         | 81.51         | 83.89   |

### C.6. RSE Case Analysis

We provide an RSE reasoning case in Figure 6. The model demonstrates a structured reasoning process by actively integrating Intermediate Conclusions to accelerate search efficiency and utilizing Failure Patterns to ensure solution validity, while exhibiting critical discernment to maintain reasoning robustness against misleading cues.

**Accelerating Reasoning Efficiency with Intermediate Conclusions.** The model leverages Intermediate Conclusions (green) as reasoning shortcuts to significantly enhance search efficiency.

- **Direct Knowledge Retrieval (Step ①):** By directly retrieving the constant $|I(S_5)| = 26$, the model substitutes a complex combinatorial sub-task with a single memory lookup. This maximizes efficiency by eliminating an error-prone manual derivation chain, allowing the limited inference budget to be focused on the

core logic.

- **Search Space Pruning (Step ②):** By enforcing the involution constraint ($f^2 = \mathrm{id}$), the model instantly reduces the candidate space from $|S_5|^3$ to $26^3$. This significantly improves reasoning efficiency by preventing the allocation of inference budget to invalid paths, allowing the model to focus its computational resources strictly on the viable subset.

- **Problem Reformulation (Step ④):** By utilizing the bijective mapping property, the model converts the recursive functional verification into a deterministic algebraic calculation ($(fg)^3 = \mathrm{id}$). This linearizes the reasoning path, effectively bypassing the high cognitive load and potential confusion associated with verifying nested recursive depths.

**Ensuring Validity and Completeness via Adversarial Auditing.** Failure Patterns (red) serve as an adversarial checklist, forcing the model to cover blind spots that standard chain-of-thought often misses.

- **Overcoming Simplicity Bias (Step ⑤):** By consulting the pattern on solution types, the model actively expands its search beyond the trivial $f = g = h$ case. This mechanism ensures the completeness of the solution set, preventing the common undercounting error where models settle for the most probable/simplest subset and ignore complex valid configurations (Type 2).

- **Global Consistency Check (Step ⑥):** Alerted by the risk of ignoring fixed points, the model extends its algebraic focus from the local 3-cycle to the full domain $S_5$. This validates the transition from local derivation to global conclusion, ensuring the structural correctness of the final answer by correctly applying the complement multiplier.

**Critical Discernment: Robustness Against Misleading Experiences.** Crucially, the model demonstrates robust reasoning by not blindly following retrieved experiences, but actively verifying their applicability to prevent being misled.

- **Active Discernment (Step ③):** Encountering the failure pattern warning against specific permutation sets, the model does not mechanically discard them. Instead, it exercises critical discernment by performing an independent calculation to verify their validity in the current context. This mechanism directly reinforces reasoning robustness, ensuring that the model remains resilient against potentially misleading retrieval cues and anchors its logical progression strictly on verified mathematical truths.

---

**Case Analysis**

**Question:** Compute the number of triples $(f, g, h)$ of permutations on $\{1, 2, 3, 4, 5\}$ such that $f(g(h(x))) = h(g(f(x))) = g(x)$, $g(h(f(x))) = f(h(g(x))) = h(x)$, and $h(f(g(x))) = g(f(h(x))) = f(x)$ for all $x \in \{1, 2, 3, 4, 5\}$.

**Answer:** 146

---

**(1) Intermediate Conclusions**

- The number of involutions in $S_5$ is 26.
- All permutations $f, g, h$ in valid solutions must be involutions, i.e., $f^2 = g^2 = h^2 = \text{id}$.
- The number of valid triples $(f, g, h)$ equals the number of ordered pairs $(f, g)$ of involutions such that $(fg)^3 = \text{id}$.

**(2) Failure Patterns**

- Failing to verify that all 6 permutations of the three transpositions on a 3-element set satisfy the equations
- Counting only Type 1 solutions (where $f = g = h$) and ignoring Type 2 solutions (where $f, g, h$ are distinct transpositions on a 3-element subset)
- Failing to account for the fixed points outside the 3-cycle support

---

**Snapshots of the Model's Reasoning Process**

...

① The number of involutions in $S_5$ is known to be 26 (1 identity, 10 transpositions, 15 double transpositions). Let's confirm... $1 + 10 + 15 = 26$. Yes, that's correct.

...

② Let's assume $f, g, h$ are involutions ($f^2 = g^2 = h^2 = e$)... The proposition 'All permutations $f, g, h$ in valid solutions must be involutions' is verified, so we can safely assume that.

...

③ The Critical Pitfalls mention that 'Failing to verify that all 6 permutations of the three transpositions on a 3-element set satisfy the equations' is a fatal flaw, but we've verified that they do, so we're safe.

...

④ The proposition 'The number of valid triples $(f, g, h)$ equals the number of ordered pairs $(f, g)$ of involutions such that $(fg)^3 = id$.' Let's see... The number of ordered pairs $(f, g)$ ... is equal to the number of Case 1 triples... plus the number of Case 2 triples...

...

⑤ The pitfall 'Counting only Type 1 solutions ($f = g = h$) and ignoring Type 2 solutions' would lead to 26, but we've counted both.

...

⑥ Another pitfall is 'Failing to account for fixed points outside the 3-element subset,' which we've addressed by multiplying by 2 (the number of involutions on the 2-element complement).

*Figure 6.* **Analysis of Reasoning Components.** The figure illustrates the problem statement, key positive constraints (Green), and critical failure modes (Red). The bottom section displays reasoning slices where the model successfully utilizes the intermediate conclusions (marked ①, ②, ④) and actively avoids the identified failure patterns (marked ③, ⑤, ⑥).

In conclusion, this case exemplifies how RSE optimizes the entire reasoning trajectory. By leveraging retrieved knowledge as both computational accelerators and adversarial auditors, the model achieves a synergy of high efficiency and rigorous completeness. Furthermore, its capacity for critical discernment ensures the reasoning process remains robust, effectively filtering noise while correctly incorporating complex truths.

# D. Prompts

We present the full prompt templates used in our experiments. Figure 7 shows the default system prompt, and Figure 8 illustrates the input serialization template for PaCoRe. Regarding our proposed RSE method, Figure 9 details the Experience Distillation prompt, while Figure 10 presents the Experience-Guided Problem-Solving prompt.

### E.1 Default System Prompt

```
Please reason step by step, and put your final answer within \boxed{}.
```

*Figure 7.* **Default System Prompt.** We apply this system instruction across all evaluated models to enforce step-by-step reasoning and standardized answer formatting.

### E.2 PaCoRe Input Serialization Template

```
You are given a problem and a list of reference responses.  Your job is to analyze these references and
provide your own response.

Original Problem:
{{ original_prompt }}

Reference Responses:
{% for response in ref_responses %}
Reference {{ loop.index }}:
{{ response }}
{% endfor %}

Now, based on the original problem and reference responses above, please provide your own comprehensive
solution.
```

*Figure 8.* **Input Serialization Template for PaCoRe.** Adopted from the PaCoRe implementation, this template embeds the current problem $x$ (denoted as original_prompt) and the reference message set $\mathcal{M}$ (denoted as ref_responses) into the model's context via Jinja2 syntax.

## E.3 Prompt for Experience Distillation.

"You are a Strategic Reasoning Distiller. Your goal is to construct a "Experience Bank" that will serve as the foundation for the student's next problem-solving iteration by extracting two specific lists:

1. **Verified Propositions:** Irrefutable truths and intermediate conclusions derived correctly.
2. **Critical Pitfalls:** Logical fallacies, dangerous operations, and dead ends to avoid.

The student will explicitly reference this data:

• Utilizing **Verified Propositions** as established anchors to accelerate valid reasoning.
• Consulting **Critical Pitfalls** to proactively avoid repeating previously identified errors, logic gaps, or dead ends.

**Constraint: strict_neutrality**
You have **NO access** to the golden answer. You must **NOT** make any assumptions about whether the student's final conclusion is correct or incorrect. Treat the student's work as an unverified hypothesis; verify the validity of each step strictly based on logic and mathematical axioms alone.

**Task 1: verified_propositions (List[str])**
**Goal:** Extract *only* mathematically sound, reusable facts (Truth Anchors).
**Strict Inclusion Rules (Filter Aggressively):**

1. **Independent Verification:** You must be able to independently verify the statement is true based on standard mathematical axioms or strictly derived from the previous valid steps.
2. **Explicit Conditions:** Every proposition MUST state its necessary conditions.
3. **Atomicity:** Break complex thoughts into the smallest reusable units.
4. **No "Lucky Guesses":** Do not include conclusions that are "likely true" but lack logical derivation.
5. **Self-Contained:** The string must be understandable without reading the original student text.

**Format:** "<Complete Statement with Conditions>. (Source: <Derivation/Method>)"

**Task 2: critical_pitfalls (List[str])**
**Goal:** Identify "Negative Constraints" that serve as warning signs for future explorations.
**Focus on identifying these specific categories:**

1. **Dead Ends (Strategy Failures):** Approaches that are technically valid but lead to unmanageable complexity or circular reasoning.
2. **Fatal Logic Flaws (Actual Errors):** Fundamental errors such as non-equivalent transformations.
3. **Potential Risks (Unsafe Operations):** Correct-looking steps that lack necessary checks (e.g., dividing by zero).
4. **Missing Proof Obligations:** Leaps in logic where a case was ignored.

**Format:** "<Context/Step> -> <Type> -> <Explanation: Trigger + Invalid Action + Consequence>"

**Output Requirements**
Output **ONLY** a raw JSON object. No Markdown formatting. Ensure all LaTeX backslashes are escaped properly.

**JSON Structure:**

```
{
    "verified_propositions": [
        "<Complete Statement>. (Source: <Derivation>)",
        "..."
    ],
    "critical_pitfalls": [
        "<Context> -> <Type> -> <Explanation>",
        "..."
    ]
}
```

**Input Data**
**Question:**
{{ question }}
**Student's Attempt:**
{{ attempt }}

*Figure 9.* Prompt for Experience Distillation.

## E.4 Experience-Guided Problem-Solving Prompt

You are an advanced mathematical solver augmented with **Experience Bank**. You are currently in a **Test-Time Scaling** loop. Previous attempts on this specific problem have been analyzed to extract useful "Propositions" (Intermediate Results) and "Critical Pitfalls" (Past Errors). Your goal is to solve the problem by starting from the definitions. Use previous memories strictly as a **navigational aid**.

**Operational Guidelines:**

**1. Accelerate via Verified Propositions (The Anchor):**
**Rule:** Treat Propositions as *structural hypotheses*, not proven facts. **Priority:** Prioritize propositions that offer abstract insights, simplifications, or identities. **Skepticism:** Be extremely skeptical of raw numerical propositions. NEVER use a specific number from the report unless you have independently derived it. **Action:** If a proposition offers a shortcut, verify its premise instantly. If valid, use it; if contradictory, discard it immediately.

**2. Navigate via Critical Pitfalls:**
The provided "Critical Pitfalls" describe specific logical errors or dead-ends. **You are STRICTLY FORBIDDEN** from repeating them. If you approach a decision point mentioned in a pitfall, you MUST actively choose an alternative strategy.

**3. Conflict Resolution & Robustness:**
**Scenario:** If you encounter a contradiction (e.g., conflicting values). **Constraint:** Do NOT simply choose the "easier" value. **Action:** A contradiction usually means a foundational assumption is incorrect. Backtrack to the very beginning, re-read the problem statement, and challenge your initial setup.

**Context from Previous Attempts:**
{{ content_of_experience_bank }}

**Instruction:**
Reason step by step. Consult the Experience Bank critically: Avoiding the previous error with pitfalls, and use propositions only if they accelerate your work. Put your final answer within \boxed{}.

*Figure 10.* Prompt for Experience-Guided Solver.

## E.5 Experience Validation Prompt

**[System Prompt]**
You are a rigorous mathematical validator. Your task is to evaluate whether each given mathematical statement is logically valid and correct in the context of the provided problem.

**Instructions:**
1. Carefully read the original problem.
2. Analyze each statement in the provided list.
3. For each statement, determine if it is mathematically correct and logically sound.
4. Output your decisions as a Python-style boolean list in the following format:
<decision>[True, False, True, ...]</decision>

**Important:**
- The list must contain exactly the same number of boolean values as the number of statements provided.
- Use True if the statement is CORRECT, False if it is INCORRECT or FLAWED.
- For propositions: Check if the intermediate result or insight is mathematically valid.
- For pitfalls: Check if the described error/pitfall is a genuine logical flaw that should be avoided.
- Be rigorous but fair in your evaluation.
- Output ONLY the <decision>[...]</decision> tag with the boolean list after your analysis.

---

**[User Template]**
**Original Problem:**
{{ problem }}

**Statement Type:** {{ statement_type }}

**Statements to Validate ({{ count }} items):**
{{ statements_list }}

Please analyze each statement and output your decisions as a boolean list with exactly {{ count }} values.
Format: <decision>[True/False, True/False, ...]</decision>

*Figure 11.* Prompt for Experience Validation.

