# OpenReview forum: "Do Not Waste Your Rollouts: Recycling Search Experience for Efficient Test-Time Scaling"
_ICML.cc/2026/Conference — Submitted to ICML 2026_

### Official Review · Reviewer_Dx7p · 2026-03-13

**Soundness:** 3
**Presentation:** 3
**Significance:** 3
**Originality:** 3
**Overall Recommendation:** 3
**Confidence:** 4

**Summary:**

This paper proposes RSE, a method for improving reasoning search efficiency in LLMs by reusing experiences gained from previous reasoning rollouts. While existing approaches sample multiple reasoning trajectories and aggregate results, the authors argue that repeated identical or similar reasoning mistakes render this search inefficient. RSE extracts intermediate conclusions, dead-end patterns, and partial insights from reasoning trajectories into an experience bank, which is then incorporated into subsequent rollout prompts to guide reasoning. Experiments on math reasoning benchmarks show improvements over self-consistency-based baselines.

**Compliance With Llm Reviewing Policy:**

Affirmed.

**Key Questions For Authors:**

**Q1**. What is RSE's core algorithmic novelty relative to recent methods such as Adaptive Branching Tree Search and RethinkMCTS, which also leverage rollout feedback to guide reasoning exploration?
**Q2**. Can the authors provide statistics on how frequently experience reuse actually occurs during the reasoning search process (e.g., reuse frequency or reuse rate per rollout)?
**Q3**. To more directly demonstrate the claimed efficiency improvement, can metrics such as expected rollouts-to-success be reported?
**Q4**. What are the individual contributions of different experience types (intermediate insights vs. dead-end patterns) to overall performance gains?
**Q5**. Do similar performance improvements hold in other reasoning domains such as coding or planning?

**Limitations:**

Yes.

**Strengths And Weaknesses:**

### Strengths
- Reusing reasoning experiences from prior rollouts to improve search is an interesting problem formulation. Separately leveraging dead-end patterns and intermediate insights is intuitive and meaningful from a reasoning trajectory analysis perspective.
- Introducing a memory-like structure into the LLM reasoning process is a simple yet practical idea.
- Empirical improvements are demonstrated across multiple reasoning benchmarks relative to trajectory sampling-based methods.

---

### Weaknesses
1. Insufficient differentiation from recent LLM reasoning search literature.
- The paper proposes reusing rollout experience to improve search efficiency, but similar ideas have been explored in recent work. For instance, Adaptive Branching Tree Search [1] proposes an adaptive search policy that decides whether to expand new reasoning branches or refine existing ones based on rollout results, while RethinkMCTS [2] uses MCTS to correct erroneous reasoning paths or generate new branches. Both approaches adjust reasoning exploration using rollout outcomes, making them conceptually related to RSE. The paper needs to more clearly articulate RSE's algorithmic novelty relative to these existing search-based reasoning methods.

2. Lack of metrics directly measuring the efficiency claim.
- Search efficiency improvement is presented as a primary contribution, yet experiments report only pass@1 or success rate. These metrics reflect accuracy gains but do not directly quantify search efficiency. Reporting metrics such as expected number of rollouts to reach a correct solution, compute-normalized performance, or sample efficiency would more convincingly support the efficiency claim. As currently presented, it is difficult to distinguish whether RSE genuinely improves search efficiency or simply improves reasoning quality.

3. Insufficient analysis of the experience reuse mechanism.
- The core mechanism that reuses reasoning experiences across rollouts lacks sufficient empirical analysis. It would be valuable to report experience reuse frequency across iterations, the individual contributions of positive insights versus dead-end patterns, and the relationship between experience bank size and performance. The existing ablations partially address experience type importance, but do not adequately explain how experience reuse operates during the actual reasoning search process.

4. Limited domain generalization.
- Experiments are conducted primarily on math reasoning benchmarks. Mathematical problems tend to have relatively clear intermediate reasoning steps and partial conclusions, which naturally favors experience reuse. Whether the same approach is effective in other reasoning domains, such as coding, planning, or knowledge-intensive reasoning, remains unverified. Additional experiments across diverse domains are needed to evaluate the method's general applicability.

---

### Reference

[1] Inoue et al., Wider or Deeper? Scaling LLM Inference-Time Compute with Adaptive Branching Tree Search, NeurIPS, 2025.

[2] Li et al., RethinkMCTS: Refining Erroneous Thoughts in Monte Carlo Tree Search for Code Generation, arXiv, 2024.

---

> ### Author Rebuttal · Authors · 2026-03-31
>
> Dear Reviewer Dx7p, thank you for your careful reading and constructive feedback. Below we respond to the concerns you raised, which will be incorporated into the revised version. Due to the space limit, for some concerns shared across reviewers, we respectfully ask the reviewer to also refer to the corresponding responses where appropriate.
>
> ### **W1&Q1. About the relation to recent MCTS-style methods.**
>
> We thank the reviewer for highlighting these highly relevant and insightful works [1][2], which are very helpful for better situating RSE within the broader inference-time search literature. We agree that [1][2] do reuse prior search information to some extent, and we will discuss them more explicitly in the revision. At the same time, it is important to clarify that, as noted in our introduction, [1][2] belong to current hybrid scaling approaches that guide search with external feedback or look-ahead evaluation. In contrast, RSE is designed for a self-contained setting and does not rely on external verifiers, execution feedback, reward models, or look-ahead evaluation. More specifically, reuse in this class of hybrid scaling methods is mainly within-branch prefix reuse, i.e., revisiting or refining a partial trajectory with branch-local feedback. RSE instead introduces cross-branch experience sharing: positive and negative experiences, such as verified facts or failure causes, are distilled from earlier rollouts and injected into later ones, so subsequent exploration can benefit from trajectory insights that would otherwise remain isolated within individual branches.
>
> [1] Wider or Deeper? Scaling LLM Inference-Time Compute with Adaptive Branching Tree Search, NeurIPS, 2025.
>
> [2] RethinkMCTS: Refining Erroneous Thoughts in Monte Carlo Tree Search for Code Generation, arXiv, 2024.
>
> ### **W2&Q3. About efficiency evaluation.**
>
> We thank the reviewer for this important question. Our paper evaluates efficiency under aligned search budgets. In the main experiments, methods are compared under the same total rollout budget, so the reported accuracy (pass@1) reflects how effectively each method uses the same amount of search. In addition, Figure 2(c) compares performance under aligned FLOPs, providing a compute-normalized view of efficiency. More broadly, in the test-time scaling community, efficiency is often assessed through task performance (e.g., accuracy) under matched rollout or compute budgets [1-3].
>
> [1] Scaling LLM Test-Time Compute Optimally Can be More Effective than Scaling Model Parameters, ICLR, 2025.
>
> [2] Inference Scaling Laws: An Empirical Analysis of Compute-Optimal Inference for LLM Problem-Solving, ICLR, 2025.
>
> [3] Large Language Monkeys: Scaling Inference Compute with Repeated Sampling, arXiv, 2024.
>
> ### **W3&Q2. About direct evidence of experience reuse during search.**
>
> We thank the reviewer for requesting a more direct analysis of how experience reuse operates during search. Since reuse in RSE is expressed textually in the reasoning process rather than through a discrete retrieval flag, we report a simple proxy based on the model’s reasoning traces. Specifically, for each reasoning content, we count the total number of occurrences of reflection, proposition, and pitfall, and then report the average over samples. As shown in the table below, these matched reuse cues are rare at Iter-0, before any experience bank is introduced, but increase sharply once the bank becomes available at Iter-1. They remain substantially above the Iter-0 level in later rounds, providing direct evidence that subsequent rollouts actively incorporate recycled experience rather than behaving like independent samples.
>
> |Avg. matched reuse cues / trace|Iter-0|Iter-1|Iter-2|Iter-3|
> |---|---:|---:|---:|---:|
> |HMMT24|6.3|655.4|139.9|119.4|
> |HMMT25|14.5|664.1|138.2|123.9|
>
> ### **Q4. About the roles of positive and negative experiences.**
>
> We thank the reviewer for this important question. This is an important aspect of RSE, and we have analyzed it in Section 4.3.3 (Synergy of Experience Components, Table 5). The results show that positive and negative experiences play complementary roles, and that using both yields the strongest performance.
>
> ### **W4&Q5. About generalization beyond mathematical reasoning.**
>
> We thank the reviewer for highlighting the importance of broader domain validation. We evaluated RSE beyond mathematical reasoning on coding (LiveCodeBench-v6), general reasoning (GPQA-Diamond), and planning (TravelPlanner). As shown below, RSE consistently outperforms strong baselines across all three domains. For task-specific implementation details, we respectfully refer the reviewer to our response to Reviewer yGMW (W3).
>
> |Benchmark|Metric|Base|Self-Ref|PaCoRe|MV@128|RSE|
> |---|---|---:|---:|---:|---:|---:|
> |LiveCodeBench-v6|pass@1|61.19|64.13|60.26|64.63|68.55|
> |GPQA-Diamond|pass@1|69.17|70.77|69.91|70.72|72.67|
> |TravelPlanner|Final Pass Rate|0.76|28.12|0.00|-|34.44|

---

> > ### Author Rebuttal · Reviewer_Dx7p · 2026-04-02
> >
> > Thank you for the thorough and responsive rebuttal. The additional experiments on coding, general reasoning, and planning (Q5) are particularly compelling, and the clarifications on Q1 and Q4 are appreciated. I have follow-up questions on Q2 and Q3:
> > - Q2: The keyword-frequency proxy is a creative approach, but it only measures whether certain words appear in the text, not whether the reused experience meaningfully contributed to reasoning. An evaluation approach such as llm-as-a-judge that assesses whether each reused experience actually influenced the reasoning trajectory would be far more convincing. Additionally, while Appendix C.6 provides a single successful case, a broader set of samples, including cases where reuse failed or was ignored, would help clarify how the mechanism operates in practice.
> >
> > - Q3: I understand that accuracy under matched budgets serves as an indirect efficiency measure. However, a more direct metric seems readily computable: for each problem, recording which rollout first produced a correct answer and reporting the average expected rollouts-to-success would directly show whether RSE reduces the number of attempts needed. This should be straightforward to derive from existing data. Would it be possible to provide this?

---

> > > ### Author Response · Authors · 2026-04-02
> > >
> > > Dear Reviewer Dx7p, thank you for the prompt and constructive follow-up. We are encouraged that our additional experiments and clarifications were helpful. Below, we respond to the remaining concerns point by point.
> > >
> > > ### Q2. About more direct evidence for meaningful experience reuse.
> > >
> > > We agree that keyword frequency is only a surface-level proxy: it measures whether reused experiences appear in the text, but not whether they meaningfully influence subsequent reasoning. Motivated by the reviewer’s suggestion, we therefore complement this proxy with a more direct LLM-as-Judge evaluation.
> > >
> > > Specifically, we sample 5 rollouts for each of the 30 questions in HMMT24, yielding 150 reasoning traces in total. For each trace, a judge model (Kimi-K2.5) examines the interaction between the provided experiences and the subsequent chain-of-thought, and classifies it into four positive-use categories: VERIFY, SHORTCUT, CORRECT, and AVOID, or three non-positive categories: REFUSE, MISUSE, and IGNORE.
> > >
> > > |Category|Avg per Rollout|
> > > |---|---:|
> > > |AVOID|2.18|
> > > |VERIFY|1.96|
> > > |SHORTCUT|1.49|
> > > |CORRECT|0.18|
> > > |Positive-use total|**5.81**|
> > > |IGNORE|0.57|
> > > |MISUSE|0.28|
> > > |REFUSE|0.09|
> > > |Non-positive-use total|**0.94**|
> > >
> > > Overall, meaningful positive use substantially outweighs non-positive interaction, with a positive-to-non-positive ratio of **6.2 : 1**. This suggests that reused experiences do not merely appear as surface text, but meaningfully shape the reasoning trajectory in most cases.
> > >
> > > Due to space limitations, we are unable to present a broader set of cases here. In the revised version, following the reviewer’s suggestion, we will include additional representative examples covering successful reuse, failed reuse, and ignored experiences.
> > >
> > > ### Q3. About reporting expected rollouts-to-success.
> > >
> > > We apologize that this point was not sufficiently clarified in our previous rebuttal. Metrics such as expected rollouts-to-success are only directly meaningful in settings where correctness can be determined using ground-truth labels, since they require identifying the first rollout that is actually correct. In the test-time scaling setting studied here, however, the method operates at test time without access to such labels, which is why mainstream test-time scaling works typically evaluate efficiency through pass@1 / accuracy under matched rollout or compute budgets, as in our main experiments.
> > >
> > > Nevertheless, we believe that the reviewer’s suggested metric can still serve as a useful offline diagnostic in more restricted settings where correctness can be assessed after generation, such as offline data synthesis, as it provides an additional view of how quickly a method reaches a correct solution under a fixed total budget. Motivated by this suggestion, we therefore additionally report a direct expected rollouts-to-success analysis. To compute this metric stably, we derive it directly from the same rollout traces, rather than from unstable one-by-one re-sampling. For each problem, we identify the smallest rollout budget k such that pass@k = 1: for the baseline, this is obtained from a large pool of parallel samples, and for RSE, from the cumulative rollouts across rounds. We exclude problems for which none of the compared methods produces any correct answer, since expected rollouts-to-success is not informative in such cases. We then report expected rollouts-to-success by averaging this minimum k over the remaining problems; if a method does not find a correct answer within the allowed budget, we assign it twice the maximum budget. Concretely, using Qwen3-30B-A3B-Thinking, we compute expected rollouts-to-success on HMMT24 and HMMT25 for different methods, summarized below:
> > > |Method|HMMT24|HMMT25|
> > > |---|---:|---:|
> > > |Base|393.92|298.43|
> > > |Self-Ref|285.03|242.82|
> > > |PaCoRe|256.42|194.61|
> > > |RSE|**216.88**|**180.21**|
> > >
> > > RSE achieves the lowest expected rollouts-to-success on both benchmarks, consistently outperforming all compared baselines.
> > >
> > > We sincerely appreciate your thoughtful suggestions. If any concern remains, we would greatly appreciate the opportunity to discuss it further, as your feedback is highly valuable for improving the clarity and quality of the paper.

---

### Official Review · Reviewer_A6TR · 2026-03-13

**Soundness:** 3
**Presentation:** 3
**Significance:** 2
**Originality:** 2
**Overall Recommendation:** 4
**Confidence:** 4

**Summary:**

The paper argues that current test-time search for LLMs wastes compute because each rollout is treated like a throwaway attempt, even when it contains useful partial progress or clear failure signals. To address this, the authors introduce Recycling Search Experience (RSE), a training-free inference method that stores and reuses information gathered during search: successful intermediate conclusions are recycled to help future rollouts reach solutions faster, while failed reasoning patterns are recycled to avoid revisiting dead ends. In this way, the search process becomes cumulative rather than memoryless. The paper also provides a theoretical analysis for why this should improve efficiency, and shows empirically on several hard reasoning benchmarks that RSE achieves better performance than strong search baselines under similar compute budgets.

**Compliance With Llm Reviewing Policy:**

Affirmed.

**Final Justification:**

Most of my concerns are resolved, so I raise my score from 3 to 4.

**Key Questions For Authors:**

1. How robust is RSE to incorrect experience labeling? Since the method relies on the model to identify positive and negative experiences, how does performance change when the model misclassifies experiences or produces unreliable self-assessments?

2. What is the computational overhead introduced by RSE? While the paper claims improved compute efficiency, could the authors provide a clearer breakdown of the additional costs of storing, retrieving, and injecting experiences during search?

3. How well does RSE generalize beyond mathematical reasoning tasks? The experiments focus primarily on math benchmarks. Have the authors evaluated or considered evaluating RSE on other reasoning domains such as coding, planning, or tool-use tasks?

4. How sensitive is the method to design choices such as memory size, deduplication strategy, and experience selection? It would be helpful to understand how stable the performance gains are under different hyperparameter settings.

**Limitations:**

It is suggested to state some limitations of RSE, like its sequential nature compared to parallel reasoning, long prompts caused by injecting experience into the prompt and so on.

**Strengths And Weaknesses:**

### Strengths

1. The paper presents a clear and intuitive idea: reusing useful reasoning experiences from previous rollouts instead of treating each rollout as independent.

2. The proposed method (RSE) is training-free, making it easy to apply to existing LLMs without additional fine-tuning. The approach is conceptually simple and practical, requiring only modifications at test-time search rather than changes to the model itself.

3. The paper provides empirical results on multiple challenging math reasoning benchmarks, demonstrating improvements over several strong baselines under similar compute budgets.

4. The experiments include useful ablations and analyses, such as examining the roles of positive and negative experiences and different design choices.

### Weaknesses

1. The evaluation is largely focused on mathematical reasoning tasks, so it is unclear how well the approach generalizes to other domains such as coding, planning, or open-ended reasoning tasks.

2. The comparisons are somewhat limited, and the paper could include a broader set of search or memory-based baselines for stronger validation.

3. The approach relies on the model’s ability to correctly identify useful and harmful experiences, which may be unreliable in some cases.

4. The paper does not fully analyze the practical overhead of storing, managing, and injecting recycled experiences during search.

5. It remains somewhat unclear which components contribute most to the observed improvements, and further analysis could clarify this.

---

> ### Author Rebuttal · Authors · 2026-03-31
>
> Dear Reviewer A6TR, thank you for your careful reading and constructive feedback. Below we respond to the concerns you raised, which will be incorporated into the revised version. Due to the space limit, for some concerns shared across reviewers, we respectfully ask the reviewer to also refer to the corresponding responses where appropriate.
>
> ### **W1&Q3. About generalization beyond mathematical reasoning.**
>
> We thank the reviewer for highlighting the importance of broader domain validation. To assess the generality of RSE beyond mathematical reasoning, we conducted additional experiments using Qwen3-30B-A3B-Thinking on coding (LiveCodeBench-v6), general reasoning (GPQA-Diamond), and planning (TravelPlanner). As shown below, RSE consistently outperforms strong baselines across all three domains. These results suggest that the benefit of RSE is not limited to math-specific settings, but more generally comes from reusing informative intermediate experience across rollouts. For task-specific implementation details, we respectfully refer the reviewer to our response to Reviewer yGMW (W3).
>
> |Benchmark|Metric|Base|Self-Ref|PaCoRe|MV@128|RSE|
> |---|---|---:|---:|---:|---:|---:|
> |LiveCodeBench-v6|pass@1|61.19|64.13|60.26|64.63|68.55|
> |GPQA-Diamond|pass@1|69.17|70.77|69.91|70.72|72.67|
> |TravelPlanner|Final Pass Rate|0.76|28.12|0.00|-|34.44|
>
> ### **W2. About broader comparisons.**
>
> We thank the reviewer for this helpful suggestion. To strengthen the empirical validation of RSE, we included additional probability- and confidence-based baselines, namely Self-Certainty (Self-C)  [1] and Deep Think with Confidence (Deep-C) [2], on HMMT24 and HMMT25 using Qwen3-30B-A3B-Thinking under the same experimental protocol as in our main experiments. As shown in the table below, RSE consistently outperforms these baselines across both datasets. These results suggest that the gains of RSE are not simply due to better answer aggregation or confidence-based compute allocation, but instead come from effectively distilling and reusing search experience across rollouts.
>
> |Dataset|Self-C@64|Self-C@128|Self-C@256|Self-C@512|Deep-C@64|Deep-C@128|Deep-C@256|Deep-C@512|RSE|
> |:--|--:|--:|--:|--:|--:|--:|--:|--:|--:|
> |HMMT24|68.3|68.1|67.8|67.3|69.3|69.3|69.6|69.7|74.4|
> |HMMT25|76.0|76.4|76.7|76.7|75.9|77.0|78.9|80.5|83.9|
>
> [1] Scalable best-of-n selection for large language models via self-certainty.
>
> [2] Deep think with confidence.
>
> ### **W3&Q1. About robustness to incorrect experience labeling.**
>
> We thank the reviewer for raising this important question. In RSE, we do not introduce any additional mechanism or external model to classify positive versus negative experiences. Instead, the model itself makes this judgment within the distillation prompt. Appendix C.5 provides a post-hoc verification of this design choice, showing validation accuracy of 81.06% on HMMT24 and 83.89% on HMMT25. We also conducted a controlled noise experiment on HMMT24, which shows that the original self-contained pipeline already distinguishes useful from harmful experiences fairly well. For the detailed setup and results, we respectfully refer the reviewer to our response to Reviewer yGMW (W1), item (1).
>
> ### **W4&Q2&L1. About computational overhead, practical cost, and the sequential-design limitation.**
>
> We thank the reviewer for highlighting this important concern. RSE does introduce additional overhead, and its round-wise design brings a latency trade-off relative to fully parallel methods. At the same time, this is closely tied to the central mechanism of RSE: later rollouts are no longer independent trials, but can benefit from distilled experience gathered in earlier rounds. Similarly, injecting recycled experiences can increase prompt length, but the resulting context-related cost is already included in the reported FLOPs (Appendix B.4). For detailed wall-clock measurements and stage-level overhead breakdown, we respectfully refer the reviewer to our response to Reviewer BY7x (W2-W3), where we report both latency comparisons and a detailed decomposition of the additional cost.
>
> ### **W5&Q4. About which components contribute most and sensitivity to design choices.**
>
> We thank the reviewer for this important point. To help clarify it, we summarize RSE from the perspective of experience reuse, which we believe is the primary source of improvement. The paper also analyzes this mechanism from several complementary angles, including semantic deduplication, information distinctness versus consistency, and the roles of positive and negative experiences in Section 4.3.3. For robustness to noisy experience labeling, we respectfully refer the reviewer to our response to Reviewer yGMW (W1), item (1). For the sensitivity to experience-bank size and rollout number, we respectfully refer the reviewer to our response to Reviewer yGMW (W1), items (2)-(3).

---

> > ### Author Rebuttal · Reviewer_A6TR · 2026-04-04
> >
> > Thank you for the detailed response. I will raise the score from 3 to 4 before the rebuttal deadline.

---

> > > ### Author Response · Authors · 2026-04-04
> > >
> > > Thank you for your prompt reply and your recognition of our work! We are glad that our rebuttal has addressed your concerns, and we will incorporate the above discussion into the final version of the paper. Thank you once again for your time and invaluable contribution to our work!

---

### Official Review · Reviewer_yGMW · 2026-03-16

**Soundness:** 2
**Presentation:** 3
**Significance:** 2
**Originality:** 2
**Overall Recommendation:** 4
**Confidence:** 3

**Summary:**

This paper proposes RSE (Recycling Search Experience), a search scaling method during inference time, to not just dispose of candidate rollouts as i.i.d components, but recycle the useful experiences or researched results via (i) Batched Experience-Guided Search, and (ii) Self-Guided Experience Distillation. The authors evaluate the effectiveness and efficiency of the method across several math reasoning tasks and conduct additional analyses, such as scaling search width, on the method.

**Compliance With Llm Reviewing Policy:**

Affirmed.

**Final Justification:**

The rebuttal mostly addresses my previous questions. I hope the authors could incorporate the new experiments to the revision of the paper. My final recommendation for this paper is weak accept.

**Key Questions For Authors:**

See the weaknesses.

**Limitations:**

yes

**Strengths And Weaknesses:**

## Strengths

The paper tackled an important problem during test-time, i.e., how to efficiently leverage different rollouts, not just as i.i.d, but distill and recycle the useful part from each candidate for more optimal information utilization.

The experiments are conducted thoroughly, including 4 benchmarks, 3 models, and several in-depth analyses to measure the efficacy of the method.

Overall, the method is simple to understand. I did not check the theoretical proofs in the Appendix, but it looks sound.

## Weaknesses

There are lots of missing details in the method description, which require additional writing and refinement to make the method description concrete. For example, how is the size of the experience bank or batch rollouts determined? Did the authors run comparison experiments to evaluate different configs? How to separate good experiences from bad experiences? What are the metrics used here for the classification problem?

The novelty of the methods seems to lie more towards the engineering experiences across recycling, rather than algorithmic innovation.

As the method claims high efficiency across different rollouts, beyond math tasks, lots of additional relevant domains should be evaluated as well. For example, models generally share similar content across different trajectories on search-based texts. It is worth seeing if the proposed method is effective across other domains as well.

Does the author apply the recycling strategy across the post-training stage rather than the inference stage? For example, in RL training, there are lots of rollouts genenrated, is the same strategy useful for training stage as well?

---

> ### Author Rebuttal · Authors · 2026-03-31
>
> Dear Reviewer yGMW, thank you for your thoughtful comments. Below we respond to the concerns you raised, which will be incorporated into the revised version.
>
> ### **W1. About method details.**
>
> (1) **About how positive and negative experiences are separated.** In RSE, we do not introduce any additional mechanism or external model to classify positive versus negative experiences. Instead, the model itself makes this judgment within the distillation prompt. Appendix C.5 provides a post-hoc verification of this design choice, showing validation accuracy of 81.06% on HMMT24 and 83.89% on HMMT25. We further conduct a controlled one-step noise experiment on HMMT24 by replacing a fraction of valid positive entries with negative ones while keeping the bank size fixed. As shown below, a fully clean bank (0% noise) improves over the original distilled bank only moderately, suggesting that the original pipeline already distinguishes useful from harmful experiences well.
>
> |Setting|Base|Original Bank|0%|20%|40%|60%|
> |---|---:|---:|---:|---:|---:|---:|
> |HMMT24 pass@1|57.4|71.9|73.8|70.1|63.5|59.5|
>
> (2) **About how the size of the experience bank is determined.** In the default RSE setting, we do not impose a hard cap on the experience bank, since deduplication already suppresses redundancy and keeps context growth manageable in practice. To study sensitivity to bank size, we use a controlled one-step setting in which the bank is constructed from the Iter-0 rollouts of RSE; when the bank exceeds the preset limit, we randomly sample experiences up to that budget for injection. The Full setting uses the original distilled bank without an explicit cap. As shown below, RSE remains competitive even with small budgets, while larger budgets bring further gains and Full performs best.
>
> |Budget|60|90|120|150|Full|
> |---|---:|---:|---:|---:|---:|
> |HMMT24|68.9|69.8|70.8|70.9|71.9|
> |HMMT25|79.2|79.0|80.6|81.1|81.3|
>
> (3) **About how the rollout number is determined.** In our main experiments, the default rollout number for RSE is set to 32 per iteration. To study sensitivity to this choice, we vary the per-iteration rollout number from 16 to 32 to 64. As shown below, performance improves as the rollout number increases. This parameter reflects a trade-off: if it is too large, more computation is spent on redundant within-batch exploration before the bank can be updated; if it is too small, sufficient search depth requires more sequential rounds and increases latency. We therefore use 32 as default.
>
> |Rollouts/iter|16|32|64|
> |---|---:|---:|---:|
> |HMMT24|71.5|74.4|74.6|
> |HMMT25|81.7|83.9|85.5|
>
> ### **W2. About whether the novelty is mainly engineering.**
>
> We agree that the current implementation of RSE is intentionally simple and lightweight, since our goal is to present a training-free method that can be easily applied to existing reasoning models. However, the main novelty of RSE does not lie merely in engineering the recycling pipeline, but in introducing a different search paradigm for test-time scaling. Existing sampling-based approaches largely treat rollouts as independent trials and only aggregate them at the answer level, whereas RSE turns search into a cumulative process by distilling reusable positive and negative experiences from earlier rollouts and injecting them into later ones. The key contribution is making rollout-level search experience-guided rather than memoryless.
>
> ### **W3. About generalization beyond mathematical reasoning.**
>
> To assess the generality of RSE beyond mathematical reasoning, we conducted additional experiments using Qwen3-30B-A3B-Thinking, extending the evaluation to coding (LiveCodeBench-v6), general reasoning (GPQA-Diamond), and planning (TravelPlanner). Since Self-Consistency relies on voting over a well-defined answer space, it is not applicable to open-ended planning tasks such as TravelPlanner. As shown below, RSE consistently outperforms all strong baselines. These results suggest that the benefit of RSE is not limited to math-specific settings, but more generally comes from reusing informative intermediate experience across rollouts.
>
> |Benchmark|Metric|Base|Self-Ref|PaCoRe|MV@128|RSE|
> |---|---|---:|---:|---:|---:|---:|
> |LiveCodeBench-v6|pass@1|61.19|64.13|60.26|64.63|68.55|
> |GPQA-Diamond|pass@1|69.17|70.77|69.91|70.72|72.67|
> |TravelPlanner|Final Pass Rate|0.76|28.12|0.00|-|34.44|
>
> ### **W4. About whether recycling can be used at the training stage.**
>
> We fully agree that extending recycling from inference-time search to rollout-intensive post-training is a valuable direction. The current paper focuses on test-time inference and does not study training-stage recycling directly. Nevertheless, in RL-style post-training or data synthesis pipelines such as rejection sampling, RSE may help reduce rollout cost by distilling earlier trajectories into reusable positive and negative experiences, so later trajectories can benefit from prior exploration instead of repeatedly searching from scratch.

---

> > ### Author Rebuttal · Reviewer_yGMW · 2026-04-03
> >
> > Thanks for the rebuttal. I have no additional questions and would like to keep my positive score for the submission.

---

> > > ### Author Response · Authors · 2026-04-04
> > >
> > > Thank you for your prompt reply and your recognition of our work! We will make sure to include the above discussion in the final version of the paper. Once again, we would like to express our heartfelt appreciation for the time and effort you dedicated to reviewing our submission. Your feedback is highly valuable to us, and we believe it will help further improve the quality of our work.

---

### Official Review · Reviewer_BY7x · 2026-03-20

**Soundness:** 2
**Presentation:** 3
**Significance:** 2
**Originality:** 3
**Overall Recommendation:** 5
**Confidence:** 4

**Summary:**

This paper proposes Recycling Search Experience (RSE), a training-free test-time scaling method that turns test-time search into a cumulative process by distilling rollouts into a shared Experience Bank of verified conclusions (positive experience) and failure patterns (negative experience), which guide subsequent search rounds. A deduplication mechanism keeps the bank compact and diverse. Experiments across challenging math benchmarks and multiple model families show RSE consistently outperforms baselines, with theoretical analysis supporting its efficiency advantage.

**Compliance With Llm Reviewing Policy:**

Affirmed.

**Final Justification:**

The authors’ rebuttal fully addressed my concerns, and I have raised my overall score accordingly.

**Key Questions For Authors:**

Please refer to the weakness section.

**Limitations:**

yes

**Strengths And Weaknesses:**

**Strengths**

- The paper is built on a well-motivated and conceptually appealing observation: the inefficiency of memoryless test-time search. The proposed RSE achieves strong empirical results against multiple baselines across diverse model families and computational budgets. Additionally, the rich analysis is insightful in deepening the understanding of RSE's behavior. Collectively, these contributions make for an informative and well-rounded paper.


**Weaknesses**

- Given that RSE maintains an experience bank to guide subsequent rollouts, a natural comparison would be a RAG-based baseline that retrieves relevant external knowledge or past solutions as context. In addition, confidence- or probability-based baselines (e.g., [1][2]) that prune low-scoring rollouts/re-weight compute allocation accordingly are well-established, training-free approaches in the test-time scaling literature that similarly require no external verifiers, yet are absent here. Including these baselines would better contextualize and strengthen RSE's empirical contributions.

- RSE's batched round structure introduces a hard sequential dependency that cannot be parallelized away: each round must fully complete before distillation runs and the next round launches. In contrast, pure parallel methods like Majority Voting complete in roughly one rollout's worth of time regardless of rollout budget, without such added latency. While the paper promotes efficiency, the analysis focuses on computational cost in FLOPs, with no wall-clock latency results -- which might be a tradeoff worth more explicit discussion.

- While RSE demonstrates consistent empirical gains, the engineering overhead introduced by the method may limit its practical applicability. For example, RSE requires an additional LLM inference pass per rollout for experience distillation and maintaining a separate text embedding model for semantic deduplication. Further ablation of leaner design choices would better justify the complexity of the full RSE framework.

--

[1] Kang et al. *"Scalable best-of-n selection for large language models via self-certainty."*

[2] Fu et al. *"Deep think with confidence."*

---

> ### Author Rebuttal · Authors · 2026-03-31
>
> Dear Reviewer BY7x, we are grateful for your careful reading and thoughtful comments. Below we respond to the concerns you raised, which will be incorporated into the revised version.
>
> ### **W1. About confidence-based and RAG-style baselines.**
>
> (1) We appreciate this valuable suggestion. While our paper primarily focuses on test-time search methods, and approaches such as Self-Consistency, Self-Certainty (Self-C) [1], and Deep Think with Confidence (Deep-C) [2] are more naturally viewed as answer-aggregation methods rather than search-based methods per se, we agree that they are still relevant comparisons, since they are also training-free and do not rely on external signals. To this end, we evaluated them on HMMT24 and HMMT25 using Qwen3-30B-A3B-Thinking under the same experimental protocol as in our main experiments. We report the best-performing tested variant for each benchmark. As shown in the table below, RSE consistently outperforms these baselines across both datasets. The results suggest that the gains of RSE are not simply due to better answer aggregation or confidence-based compute allocation, but instead come from effectively distilling and reusing search experience across rollouts.
>
> |Dataset|Self-C@64|Self-C@128|Self-C@256|Self-C@512|Deep-C@64|Deep-C@128|Deep-C@256|Deep-C@512|RSE|
> |:--|--:|--:|--:|--:|--:|--:|--:|--:|--:|
> |HMMT24|68.3|68.1|67.8|67.3|69.3|69.3|69.6|69.7|74.4|
> |HMMT25|76.0|76.4|76.7|76.7|75.9|77.0|78.9|80.5|83.9|
>
> (2) Regarding RAG-style baselines, we note that external RAG addresses a somewhat different setting from the one studied in this paper: RAG augments reasoning with externally retrieved knowledge or past solutions, whereas RSE focuses on online, within-instance reuse of search experience generated during the current inference process. Moreover, adapting RAG to our setting would require additional design choices, since existing RAG methods are typically designed for clean external knowledge rather than long, noisy reasoning traces.
>
> [1] Scalable best-of-n selection for large language models via self-certainty.
>
> [2] Deep think with confidence.
>
> ### **W2. About sequential dependency and missing wall-clock analysis.**
>
> We thank the reviewer for this constructive suggestion. To address this concern, we have added wall-clock latency (seconds) measurements in the table below. We agree that RSE incurs higher elapsed time than Self-Consistency, since each round must wait for experience distillation and bank update before the next round can begin. This is a real trade-off of the sequential design. In addition, injected experience increases rollout cost through longer context, and this overhead is already included in the rollout FLOPs (Appendix B.4). At the same time, our main claim is not to achieve the minimum elapsed time under abundant parallel resources, but to improve test-time search by converting previous rollouts into reusable guidance for subsequent sampling, rather than treating each rollout as an independent trial. In practical settings, available parallelism, batch size, and GPU memory are all finite, so efficiency in test-time scaling is often discussed under compute budget (e.g., FLOPs) [1-2]. From this perspective, the additional sequential cost of RSE is precisely what enables later rollouts to become more informed and reduces redundant exploration.
>
> |Method|MV@128|PaCoRe|RSE|Self-Ref|
> |---|---:|---:|---:|---:|
> |HMMT24|365.93|732.66|1200.99|1382.28|
> |HMMT25|347.43|631.52|1052.16|1248.26|
>
> [1] Scaling LLM Test-Time Compute Optimally Can be More Effective than Scaling Model Parameters
>
> [2] Inference Scaling Laws: An Empirical Analysis of Compute-Optimal Inference for LLM Problem-Solving
>
> ### **W3. About engineering overhead and practical applicability.**
>
> We agree that a clearer cost breakdown is important. Appendix B.4 provides a detailed computational cost estimation for RSE by decomposing its token consumption into the distillation and reasoning phases. To complement that analysis, we further present a stage-level breakdown here. The dominant additional cost comes from the extra experience distillation stage after each round, together with the sequential dependency between rounds. By contrast, experience-bank management itself is relatively lightweight: after distillation and semantic deduplication, the bank remains compact, so storing, sampling, and injecting experiences add limited cost compared with the main LLM stages. Our measurements also show that semantic deduplication contributes negligible FLOPs relative to LLM decoding. Overall, the main overhead of RSE comes from the additional LLM pass and serialized scheduling, rather than bank management itself.
>
> |Stage|PFLOPs|latency/s|
> |---|---:|---:|
> |Iter-0 Rollout|8.0|363.8|
> |Iter-1 Reflect|9.9|78.3|
> |Iter-1 Dedup|9.8e-5|24.0|
> |Iter-1 Rollout|6.6|198.2|
> |Iter-2 Reflect|5.1|96.3|
> |Iter-2 Dedup|1.9e-4|31.0|
> |Iter-2 Rollout|8.4|190.1|
> |Iter-3 Reflect|4.9|92.9|
> |Iter-3 Dedup|2.8e-4|34.0|
> |Iter-3 Rollout|7.9|181.3|

---

> > ### Author Rebuttal · Reviewer_BY7x · 2026-04-01
> >
> > Thanks for the comprehensive response. My concerns have been adequately addressed and I've updated my score accordingly.

---

> > > ### Author Response · Authors · 2026-04-01
> > >
> > > Thank you for your prompt reply and your recognition of our work! We will make sure to incorporate the above discussion into the final version of our paper. Once again, we sincerely appreciate the time and effort you dedicated to reviewing our submission. Your suggestions are truly valuable, and we believe they will further improve the quality of our work.

---

### Decision · Program_Chairs · 2026-04-30

**Decision:**

Reject

**Comment:**

This paper studies test-time scaling for LLM reasoning. The proposed method improves search by recycling useful positive and negative experiences from earlier rollouts into later ones, turning test-time search from a memoryless process into a cumulative one.

Reviewers agreed that the paper studies an important problem, and several appreciated the core idea, the training-free design, and the empirical results across multiple benchmarks.

However, the major concern is that the current version does not establish a clear distinction from prior work. The proposed method is more naturally viewed as a new form of feedback-guided search in which the model generates and reuses its own textual feedback, rather than as a fundamentally different “self-contained” search paradigm as they claimed. In addition, the current version does not sufficiently compare or position itself against the relevant literature on feedback-guided search. Although the rebuttal added more experiments and analyses, it still does not fully resolve whether the contribution should be understood as a genuinely new search formulation or as a specific variant within the broader feedback-guided search literature.

Overall, while the paper contains promising ideas and strong empirical results, the current version would benefit from a clearer and more accurate positioning of its novelty before it is ready for publication at ICML.